



# Examining the Role of Environmental Memory in the Predictability of Carbon and Water Fluxes Across Australian Ecosystems

Jon Cranko Page[1,2], Martin G. De Kauwe[3,1,2], Gab Abramowitz[1,2], Jamie Cleverly[4], Nina Hinko-Najera[5], Mark J. Hovenden[6], Yao Liu[7], Andy J. Pitman[1,2], and Kiona Ogle[8]

[1]ARC Centre of Excellence for Climate Extremes, Sydney, NSW 2052, Australia
[2]Climate Change Research Centre, University of New South Wales, Sydney, NSW 2052, Australia
[3]School of Biological Sciences, University of Bristol, 24 Tyndall Avenue, Bristol, BS8 1TQ, UK
[4]Terrestrial Ecosystem Research Network, College of Science and Engineering, James Cook University, Cairns, QLD 4870, Australia
[5]School of Ecosystem and Forest Sciences, The University of Melbourne, 4 Water Street, Creswick, VIC 3363, Australia
[6]Biological Sciences, School of Natural Sciences, University of Tasmania, Hobart, TAS 7005, Australia
[7]Department of Geography and Environmental Sciences, Northumbria University, Newcastle upon Tyne, NE1 8ST, UK
[8]School of Informatics, Computing, and Cyber Systems, Northern Arizona University, Flagstaff, Arizona, 86011, U.S.A.

**Correspondence:** Jon Cranko Page (joncrankopage@gmail.com)

**Abstract.** The vegetation's response to climate change is a significant source of uncertainty in future terrestrial biosphere model projections. Constraining climate-carbon cycle feedbacks requires improving our understanding of the direct, as well as the long-term plant physiological responses to climate. In particular, the timescales and strength of memory effects arising from both extreme events (i.e. droughts and heatwaves) and structural lags in the systems have largely been overlooked in the

development of models.This is despite the knowledge that plant responses to climatic drivers occur across multiple timescales (seconds to decades), with the impact of climate extremes resonating for many years.

Using data from 13 eddy covariance sites, covering two rainfall gradients (256 to 1491 mm yr$^{-1}$) in Australia, in combination with a hierarchical Bayesian model, we characterised the timescales and magnitude of influence of antecedent drivers on daily net ecosystem exchange (NEE) and latent heat flux ($\lambda$E). Model fit varied considerably across sites when modelling NEE, with

$R^2$ values of between 0.30 and 0.83. Latent heat was considerably more predictable across sites, with $R^2$ values ranging from 0.56 to 0.95. When considered at a continental scale, both fluxes were more predictable when memory effects were included in the model. These memory effects accounted for an average of 17% of the NEE predictability and 15% for $\lambda$E. The importance of environmental memory in predicting fluxes increased as site water availability declined ($\rho$ = -0.72, p < 0.01 for NEE, $\rho$ = -0.62, p < 0.05 for $\lambda$E). However, these relationships did not necessarily hold when sites were grouped by vegetation type.

We also tested a k-means clustering plus regression model to confirm the suitability of the Bayesian model for modelling these sites. The k-means approach performed similarly to the Bayesian model in terms of model fit, demonstrating the robustness of the Bayesian framework for exploring the role of environmental memory. Our results underline the importance of capturing memory effects in models used to project future responses to climate change, especially in water-limited ecosystems. Finally, we demonstrate a considerable variation in individual site predictability, driven to a notable degree by environmental memory,

and this should be considered when evaluating model performance across ecosystems.



# 1 Introduction

Ecosystems respond to climate at a wide range of timescales: stomata respond to changes in humidity within seconds (Fanjul and Jones, 1982), while extreme droughts can impact plant growth for up to five or more years after the event (Anderegg et al.,
2015; Vanoni et al., 2016). There is growing interest in better understanding these timescales over which vegetation responds to environmental conditions, particularly when considering the projected rate of future climate change and threats this poses to ecosystem stability (Mottl et al., 2021). An increasing number of studies have demonstrated the importance of past events such as fires (Sun et al., 2020), land management (Seabloom et al., 2020) and droughts (Anderegg et al., 2015; Kannenberg et al., 2020)) on current ecosystem behaviour, in many cases, over timespans of years.

It is important to differentiate between what shall be referred to as "legacy" and "lag" effects. The ongoing impacts of climate extremes are an example of the former - they leave a persistent yet diminishing "legacy" from their single occurrence. A "lagged" effect differs, in that it is an ongoing, constant delay in reaction to current conditions. The differing response times of vegetation to stimuli mean that lags exist associated with climate and the exchange of heat, energy and carbon fluxes between ecosystems and the atmosphere. For instance, current grassland soil respiration can be strongly influenced by antecedent
moisture from the prior two weeks, with the importance of longer lags decreasing sharply despite cumulative lag effects showing for up to six weeks (Cable et al., 2013). These response timescales are affected by ecosystem characteristics, such as vegetation structure. Cable et al. (2013) also found that, relative to grasslands, shrublands had a much longer cumulative lag effect in soil respiration of up to ten weeks, with the first four weeks being most important.

The timescale of responses also differ among ecosystem processes. Feldman et al. (2020) found a lag of five days between
a pulse of rainfall and peak plant water content in semi-arid grasslands, which in turn is likely to affect plant water status and hence carbon uptake. In arid and semi-arid ecosystems, soil respiration can respond immediately to a rainfall pulse and remain elevated for up to 2 days, while NEE rates have a lagged response of up to a week (Huxman et al., 2004; Cleverly et al., 2013). Cleverly et al. (2016) found a variety of lags for phenological and photosynthetic responses in central Australia, ranging from immediate to six weeks. Antecedent climate is yet another factor that influences ecosystem response timescales. Repeated
droughts in one growing season can negatively affect a plant's investment in biomass in comparison to a single event (Lemoine et al., 2018). In semi-arid and arid regions, NDVI (Normalised Difference Vegetation Index, a measure of vegetation greenness) responds to precipitation with lags of up to 7 months, while less arid areas tend to have shorter response times (Liu et al., 2018). From a study over the entire state of Kansas, Wang et al. (2003) found shorter lags between NDVI and precipitation, with the strongest correlation being at a 4-week lag. In fact, antecedent conditions could even be more important than concurrent
conditions when measuring ecosystem productivity (Sala et al., 2012). Clearly, there is a diverse range of lagged responses to climate found within terrestrial ecosystems. Confounding factors, such as vegetation type, prevailing climate, interacting processes, and prior extreme events, all influence the magnitude and timescale of these lags and their impact on ecosystem fluxes.





Despite increasing attention on the impact of these lags within ecosystems, they are poorly captured within Terrestrial
Biosphere Models (TBMs) (Anderegg et al., 2015; Frank et al., 2015; Ogle and Barber, 2016; Kannenberg et al., 2020).
For example, TBMs usually have an instantaneous coupling between photosynthetic uptake and growth in plants, while in
reality carbon is first allocated to storage (non-structural carbohydrates), allowing it to sustain growth and respiratory demands
during periods of lower photosynthetic activity (Fatichi et al., 2014; Smith and Dukes, 2013; Jones et al., 2020). Anderegg et al.
(2015) similarly found that Earth System Models are generally weak at predicting the impact of droughts on productivity during
the drought recovery period. Models generally overestimate the immediate impact and underestimate the recovery times for
extreme events (Kolus et al., 2019; Huang et al., 2016; Ukkola et al., 2016a). Recent satellite observations have also indicated
that models fail to capture the impact of water stored in reservoirs with longer response times to climate (such as deeper soil
moisture, groundwater and surface water), resulting in models being more dominated by anomalies at shorter timescales than
observations (Humphrey et al., 2018). Implementing these"lagged effects" into the models used to predict the carbon cycle is
key to improving their performance (Keenan et al., 2012).

To better incorporate the role of lags in model process representations, we first need to determine which physical processes
are affected and then quantify the timescales at which lags persist. Frequently, auto-regressive methods are used in studies
looking at specific pre-determined timescales, such as the current year's productivity and/or the previous year's rainfall (Sala
et al., 2012), or the prior six months of climate (Zhang et al., 2015). Many studies have however indicated that intra-annual
rainfall patterns are more important to productivity than total annual rainfall (Hovenden et al., 2014, 2018). Overcoming these
subjective constraints and still providing freedom to explore sensitivity to shorter (sub-yearly) lags seems key to revealing
any unexpected behaviour and detailing the full extent of memory effects, including any potential interaction between climate
drivers.

To address this issue of strictly prescribed lags, new statistical approaches have been developed that allow for more flexible
estimation of timescales of influences and magnitudes of lag effects (Ogle et al., 2015). Such methods have identified a key
role of memory effects in many ecosystem processes, including soil and ecosystem respiration (Ryan et al., 2015; Cable et al.,
2013; Barron-Gafford et al., 2014) and gross primary production(Ryan et al., 2017). Net ecosystem exchange (NEE) and latent
heat flux ($\lambda$E) have also been shown to be influenced by these memory effects (Samuels-Crow et al., 2020), which are often
stronger at drier sites (Liu et al., 2019).

Here we use the stochastic antecedent modelling (SAM) framework of Liu et al. (2019) to probe lagged flux responses in
individual ecosystems across two environmental gradients in Australia. We explore whether a relationship between site arid-
ity and importance of antecedent conditions holds when viewed at a smaller spatial scale, or whether such a relationship is
confounded by other site characteristics, such as vegetation structure or extreme weather conditions. By focusing on a number
of intensively-studied sites within a single continent, and examining the relative contributions of various predictors in each
environment, we aim to reduce these potential confounding factors. Our second aim is to explore whether assessing the im-
portance of environmental memory at a site can offer insights into TBM evaluation. If sites are more predictable, or have a
greater dependence on antecedent conditions (and therefore are traditionally less predictable), this should be taken into account
when critically analysing TBM performance across sites. Finally, we examine an alternative statistical approach for antecedent



| Site Name | Site Code | MAP (mm) | CVP (%) | WI | Years Analysed | Reference |
|---|---|---|---|---|---|---|
| Alice Springs Mulga* | AU-ASM | 321 | 61 | 0.12 | 2011 - 2018 | Cleverly and Eamus (2015a) |
| Calperum | AU-Cpr | 256 | 17 | 0.12 | 2011 - 2019 | Koerber (2015) |
| Cumberland Plains | AU-Cum | 902 | 34 | 0.52 | 2013 - 2019 | Pendall (2015) |
| Daly River Uncleared* | AU-DaS | 1130 | 107 | 0.48 | 2008 - 2018 | Beringer and Hutley (2015a) |
| Dry River* | AU-Dry | 842 | 112 | 0.34 | 2009 - 2019 | Beringer and Hutley (2015b) |
| Gingin | AU-Gin | 696 | 83 | 0.33 | 2012 - 2019 | Silberstein (2015) |
| Great Western Woodlands | AU-GWW | 273 | 30 | 0.11 | 2013 - 2019 | Macfarlane (2016) |
| Howard Springs* | AU-How | 1486 | 104 | 0.75 | 2002 - 2019 | Beringer and Hutley (2015c) |
| Sturt Plains* | AU-Stp | 573 | 107 | 0.23 | 2009 - 2019 | Beringer and Hutley (2015d) |
| Ti Tree East* | AU-TTE | 312 | 61 | 0.09 | 2013 - 2018 | Cleverly and Eamus (2015b) |
| Tumbarumba | AU-Tum | 1491 | 32 | 1.20 | 2002 - 2018 | Woodgate (2015) |
| Whroo | AU-Whr | 577 | 20 | 0.36 | 2012 - 2018 | Beringer and Hutley (2015e) |
| Wombat | AU-Wom | 1069 | 33 | 0.84 | 2011 - 2019 | Arndt (2013) |

**Table 1.** Site information for all sites included in the analysis. MAP is the mean annual precipitation at the site. CVP is the coefficient of variation of precipitation. WI is a wetness index of MAP over mean annual reference evapotranspiration. MAP CVP, and WI are calculated from the WorldClim dataset, which covers 1970-2000 (Fick and Hijmans, 2017; Trabucco and Zomer, 2018). Years Active corresponds to the full years of data available for analysis, inclusive. Asterisks indicate sites that are part of the NATT. Litchfield was initially included in the analysis but was discarded due to the short time series available. All other sites are grouped as the SAWS. Data were obtained from the OzFlux Data Portal (http://data.ozflux.org.au).

modelling to test whether structural assumptions in the SAM framework notably affect inferences about environmental lags. In
addition to highlighting sites with more complex lagged environmental behaviour (a tougher test for TBMs), explicit identification of individual lagged site mechanisms can highlight key processes missing from TBMs more broadly.

## 2 Methods

### 2.1 Datasets

#### 2.1.1 Flux data

Meteorological and flux data were taken from the OzFlux data repository for 13 eddy covariance towers (see Table 1 for site details). The sites were selected to cover a variety of vegetation types and fall into two overarching groups. Firstly, sites comprising the North Australia Tropical Transect (NATT) were included. These vary from tropical grasslands to semi-arid shrublands and savannahs along a steep rainfall gradient (312 to 1486 mm annual precipitation) running from north to south Australia. Secondly, we grouped the Southern Australian Woodland Sites (SAWS). These were selected as sites with a greater





proportion of woody vegetation than the NATT sites, while still covering a broad range of climate types (256 to 1491 mm annual precipitation). While the vegetation differences between the groups are less distinguished at the drier sites, the NATT sites are considered "savannah" sites, while the SAWS sites will be referred to as "woodland" sites.

     In this analysis we used net ecosystem exchange (NEE) and latent heat ($\lambda$E) fluxes. NEE is a direct measurement of carbon exchange which represents the balance of carbon uptake and losses, and is favoured over flux-derived gross primary produc-

tivity due to issues with respiration partitioning (Renchon et al., 2021). $\lambda$E measures all evaporation, including contributions from soil and vegetation. Although carbon (uptake) and water fluxes are non-linearly coupled (De Kauwe et al., 2015a), NEE and $\lambda$E are expected to exhibit a degree of independence in their responses to environmental conditions (including differing timescales), since they contain contributions from different components of the system (e.g. soil/understorey vs canopy).

     The fluxes were predicted using meteorological observations which included mean downward shortwave radiation, mean

air temperature, mean vapour pressure deficit (VPD), and precipitation. All OzFlux data were extracted at a daily timestep, screened to only include complete calendar years and then mean-centred. Sites were screened to ensure that at least five years of good quality data were available, which excluded four sites that have previously been included in the NATT (Adelaide River, Daly River Pasture, Fogg Dam, and Litchfield). Details on the data processing and quality control of OzFlux data can be found in Isaac et al. (2017). Here, we used the L6 data which is quality checked and gap-filled.

Normalized Difference Vegetation Index (NDVI), a proxy for vegetation greenness, was obtained for each site from the global MODIS Daily Albedo data product at a 500 m resolution via the Google Earth Engine interface (Schaaf and Wang, 2015; Gorelick et al., 2017). This is used in the Bayesian model as a measure of whether the sites were in a "growing" (higher greenness) or "dormant" (lower greenness) state (see below).

     General site characteristics including the mean annual precipitation (MAP) and the coefficient of variation of precipitation

(CVP) were obtained from the WorldClim dataset (Fick and Hijmans, 2017), while the wetness index (WI) was calculated from these data as MAP divided by mean annual potential evapotranspiration (Trabucco and Zomer, 2018). This WI assumes that lower values indicate more water-limited locations while higher values are locations with a greater proportion of precipitation to potential evaporation. These data were used in preference to meteorological data from individual sites as they cover a significantly long time period.

## 2.2   Analysis

### 2.2.1   Statistical Analysis of Memory and Lags

Following previous work by Liu et al. (2019) in the application of the SAM framework, we separately model NEE and $\lambda$E using a nonlinear mixed-effects Bayesian model at each site. The same formulation was used to model both fluxes at the 13 sites, and so the following description of the model is also applicable to $\lambda$E modelling.





Daily NEE is assumed to be Laplace-distributed (Richardson et al., 2006) with mean $\mu NEE$ and variance $\sigma^2$. $\mu NEE$ at time $t$ is modelled as per Equation 1.

$$\mu NEE(t) = \sum_{n=1}^{16} \left( \left[ \phi(t) \times G_n + \left( 1 - \phi(t) \right) \times D_n \right] \times CLIMATE_n(t) \right) \tag{1}$$

$G_n$ and $D_n$ represent two sets of coefficients, corresponding to "Growing" and "Dormant" behaviours respectively. These two behaviours are a function of NDVI, which we used to capture a site's growing seasons. Growing season differences were

accounted for in our statistical modelling because at many of the sites growth is restricted by water availability (e.g., following rain), which in Australia is not strictly related to a specific growth period during the year. $\phi(t)$ is a function that partitions between these two behaviours based on NDVI, such that $D_n$ is 1 and $G_n$ is 0 at minimum NDVI and vice versa at maximum NDVI:

$$\phi(t) = \left[ 1 - \phi_* + \phi_* \times NDVI(t) \right] \times NDVI(t) \tag{2}$$

$\phi_*$ is assigned a uniform prior on the interval [-1,1]. This ensures that $\phi(t)$ increases monotonically with increasing NDVI. The $\phi_*$ value modifies this function, allowing either a one-to-one linear relationship (when $\phi_*$ is equal to 0) or varying non-linear relationships. In this manner, the proportion of timesteps assigned to predominately "growing" or "dormant" behaviour (when $\phi(t)$ takes a value greater than 0.5, or less than 0.5 respectively) is modified during the model fitting.

$CLIMATE_n(t)$ is a weighted sum of climate observations with various lags, where $n$ indicates various variables as follows.

When $n = 1$, $CLIMATE_n(t) = 1$ at all timesteps $t$ and is therefore an intercept term. Where $n = 2,3,4,5$, $CLIMATE_n(t)$ represents the contribution of each of the short-term climate predictors (e.g. shortwave radiation, air temperature, etc). $n = 6,...,9$ are the quadratic interactions of these predictors and $n = 10,...,15$ are the remaining pairwise interactions. $n = 16$ is the contribution of long-term precipitation. Equation 3 is the generic form for a short-term climate predictor where $\omega_{lag}^{CLIM_n}$ are the weights assigned to the lagged climate observations $CLIM_n(t-lag)$. The $CLIMATE_n(t)$ terms for each $n$ are summarised

in Table 2.

Long-term precipitation (up to 365 days prior) is included as it is possible for the vegetation at these sites to be drawing on water reservoirs from deeper within the soil profile than would be recharged by short-term precipitation. This is especially true for the monsoonal sites along the NATT which experience prolonged dry seasons. In comparison, radiation, temperature, and VPD are assumed not to influence the fluxes at timespans longer than 14 days, and so are restricted to the shorter timescale

(Ryan et al., 2017).

$$CLIMATE_n(t) = \sum_{lag=0}^{13} \left[ \omega_{lag}^{CLIM_n} \times CLIM_n(t-lag) \right] \tag{3}$$





Each $CLIMATE_n(t)$ term has a unique set of weights, which were assigned a Dirichlet prior. This ensures that each individual weight is constrained between 0 and 1, and the sum of the weights within each $CLIMATE_n(t)$ term is equal to 1. As such, each weight is indicative of the relative importance of the specific climate variable at the corresponding lag period.

To reduce the number of parameters estimated in the model, the lagged time steps in the short-term $CLIMATE_n(t)$ sums were grouped into blocks, with each lag in a block assigned the same weight, as shown in Table 3. For instance, the same weight is assigned to observations from 7 and 8 days prior. The decreasing resolution further into the past is due to the expectation that the importance of the driving variable on consecutive days becomes increasingly difficult to distinguish as the lag time increases. This reduction in the number of individual weights being estimated improves the model computation time and

convergence.

     Significant changes between Liu et al. (2019) and our model include the removal of soil moisture, the introduction of shorter precipitation lags ($< 14$ days), and min-max normalisation of NDVI on a per-site basis. We excluded soil moisture from our analysis due to inconsistencies in the data record at some sites, as well as varying measurement depths across sites. Additionally, the soil moisture data are typically collected from relatively shallow depths of the soil profile and, as such, may

not accurately reflect the root zone soil moisture in woody ecosystems. For instance, eucalyptus species are known to have dimorphic rooting profiles with deep tap roots able to access water from below the shallow root zone (Knight, 1999). To account for short-term impacts of water availability following the removal of soil moisture, precipitation lags at the shorter timescales were introduced. Finally, NDVI is normalised to vary from 0 at minimum NDVI to 1 at maximum observed NDVI at each site for two reasons. Firstly, this ensures that each site spans the full range of $\phi$ values between "growing" (1) and

"dormant" (0) periods. The similarity between the individual $G_n$ and $D_n$ coefficients was then able to indicate any differences between these two behaviours. Secondly, it was hypothesised that totally excluding coefficients of one behaviour at minimum NDVI (and the other at maximum NDVI) would improve convergence towards well-defined parameter values during model fitting.

     The Bayesian model was implemented in JAGS via R, using the r2jags package and Markov chain Monte Carlo (MCMC)

simulations (Plummer, 2003; R Core Team, 2020; Su and Yajima, 2020). Convergence of the MCMC chains was confirmed visually with trace plots, and analytically with the Gelman-Rubin and Geweke diagnostic values as calculated with the *coda* package (Plummer et al., 2006). The MCMC iterations were set high enough to minimise the number of parameters with an "effective sample size" of less than 10,000, which ensured the parameters' posterior distributions were sufficiently sampled (Kruschke, 2015; Harms and Roebroeck, 2018).

Once converged, model performance was assessed using five metrics: coefficient of determination ($R^2$), correlation coefficient (CCO), standard deviation difference (SDD), mean bias error (MBE), and normalised mean error (NME) (Haughton et al., 2016). These were calculated between the daily time series of the observed and modelled flux. Together, these metrics capture a broad range of potential model performance measures relative to the observations.

     The model was first run in a "current climate-only" configuration, referred to as the "CC model", where the weights were set

to 0 for all but the current day climate (which were therefore assigned a weight of 1). This restriction was removed for a second set of model runs to utilise the full SAM framework. This second set of runs introduces "environmental memory" to the flux



| | Climate Predictor | Formulation |
|---|---|---|
| | $(n)$ | $(CLIMATE_n(t))$ |
| Intercept Term | 1 | $= 1$ |
| Short-term Predictors | 2 | $= TAS_{ant}(t)$ <br> $= \sum_{lag=0}^{13} \left[ \omega_{lag}^{TAS} \times TAS(t-lag) \right]$ |
| | 3 | $= SWR_{ant}(t)$ <br> $= \sum_{lag=0}^{13} \left[ \omega_{lag}^{SWR} \times SWR(t-lag) \right]$ |
| | 4 | $= VPD_{ant}(t)$ <br> $= \sum_{lag=0}^{13} \left[ \omega_{lag}^{VPD} \times VPD(t-lag) \right]$ |
| | 5 | $= PPT_{ant}^{short}(t)$ <br> $= \sum_{lag=0}^{13} \left[ \omega_{lag}^{PPT^{short}} \times PPT(t-lag) \right]$ |
| Quadratic Terms | 6 | $= TAS_{ant}(t) \times TAS_{ant}(t)$ |
| | 7 | $= SWR_{ant}(t) \times SWR_{ant}(t)$ |
| | 8 | $= VPD_{ant}(t) \times VPD_{ant}(t)$ |
| | 9 | $= PPT_{ant}^{short}(t) \times PPT_{ant}^{short}(t)$ |
| Pairwise Interactions | 10 | $= TAS_{ant}(t) \times SWR_{ant}(t)$ |
| | 11 | $= TAS_{ant}(t) \times VPD_{ant}(t)$ |
| | 12 | $= TAS_{ant}(t) \times PPT_{ant}^{short}(t)$ |
| | 13 | $= SWR_{ant}(t) \times VPD_{ant}(t)$ |
| | 14 | $= SWR_{ant}(t) \times PPT_{ant}^{short}(t)$ |
| | 15 | $= VPD_{ant}(t) \times PPT_{ant}^{short}(t)$ |
| Long-term Predictor | 16 | $= PPT_{ant}^{long}(t)$ <br> $= \sum_{lag=13}^{365} \left[ \omega_{lag}^{PPT^{long}} \times PPT(t-lag) \right]$ |

**Table 2.** Formulas for the $CLIMATE_n(t)$ term in the model for $\mu NEE(t)$ for each $n$. $TAS$ is mean air temperature, $SWR$ is incoming short-wave radiation, $VPD$ is vapour pressure deficit, and $PPT$ is rainfall. The lag term in the sums is in days i.e. at lag = 3, $TAS(t-lag)$ is the mean air temperature from 3 days prior. Note, the lags in $PPT_{ant}^{long}$ are larger periods than the daily lags for the short-term predictors, and $PPT$ is here taken as the mean daily rainfall in these lagged periods. $\omega_{lag}^{CLIM}$ is the weight assigned to each lag period and is different for each of the 5 climate variables.

predictions, and so this model configuration is referred to as the "EM model". Finally, in an attempt to capture any remaining predictability in the observations, the residuals from the EM models were themselves modelled using a standard autoregressive process with a time lag of one day, which is an AR(1) model. This effectively allows a correction of the predicted flux based



| Weight | Short-term | Long-term |
|:------:|:----------:|:---------:|
| 1 | $t$ | $t-14$ to $t-20$ |
| 2 | $t-1$ | $t-21$ to $t-29$ |
| 3 | $t-2$ | $t-30$ to $t-59$ |
| 4 | $t-3$ | $t-60$ to $t-119$ |
| 5 | $t-4$ | $t-120$ to $t-179$ |
| 6 | $t-5$ | $t-180$ to $t-269$ |
| 7 | $t-6$ | $t-270$ to $t-365$ |
| 8 | $t-7, t-8$ | **NA** |
| 9 | $t-9, t-10$ | **NA** |
| 10 | $t-11, t-12, t-13$ | **NA** |

**Table 3.** Assignment of weights to lagged periods. The same weight is applied to multiple lag periods for the short-term predictors. Grouping of lags is necessary to reduce the number of parameters being estimated by the Bayesian framework, which improves model computing performance and reduces the risk of overfitting.

on the prior error in the prediction. While previously this has been referred to as capturing a "biological memory" component, there are various plausible effects that this AR(1) model can be considered to represent (Liu et al., 2019). As such we do not claim it represents a specific memory process but rather an lower bound on site predictability, and it is simply referred to as the "AR model".

### 2.2.2 Additional Modelling

For comparison and confirmation of the SAM method, site NEE and $\lambda E$ were also modelled using a k-means clustering plus regression approach (Abramowitz, 2012; Best et al., 2015). The k-means approach is an alternative in-sample empirical model, providing a direct comparison to the SAM approach. The clustering is performed on the environmental predictor variables and the time steps that belong within each cluster are determined. For each cluster, a linear regression between the climate predictors and the flux at the time steps within the cluster is performed. This allows assessment of the degree to which SAM results are

indicative of site behaviour as opposed to resulting from the inherent structure of the SAM model. Seven different cluster plus regression models were implemented. Firstly, fluxes were modelled as a linear combination of concurrent-only climate variables. Five additional models were then run, where each model included concurrent climate but was further expanded with one individual climate predictor including potential lags (see Table 3 for lag timescales for each of the predictors). Finally, clustering and regression was performed for a model including all concurrent and lagged climate variables. As such, this final

model contains the same information as that of the EM SAM model. The NbClust package (Charrad et al., 2014) indicated that for the majority of sites and models, four or less clusters were preferred for model parsimony. While increasing the number of clusters would increase the $R^2$ values reported, we found that clusters began to contain less than a reasonable number of observations for the linear regression (less than 260 observations, which is four times the number of parameters in the k-means





model containing every lag). As such, we repeated the k-means clustering for each site and model, with the number of clusters
       ranging from two to eight, and the median $R^2$ value taken as the measure of performance.

       For further comparison, we also consider the performance of an uncalibrated TBM in simulating site NEE. The TBM used
       was the CSIRO Atmosphere Biosphere Land Exchange (CABLE) model (Kowalczyk et al., 2006), a land surface scheme that
       can be run offline with prescribed meteorological forcing (De Kauwe et al., 2015b; Decker et al., 2017; Haverd et al., 2018;
       Ukkola et al., 2016b; Wang et al., 2011), or fully coupled (Lorenz et al., 2014; Pitman et al., 2011) within the Australian

Community Climate Earth System Simulator (ACCESS; Kowalczyk et al. 2013). CABLE models the exchange of carbon,
       energy and water fluxes at the land surface, representing the vegetation with a single layer, two- leaf (sunlit/shaded) canopy
       model (Wang and Leuning, 1998) and a detailed treatment of within-canopy turbulence (Raupach, 1994; Raupach et al., 1997).
       Soil water and heat conduction are numerically integrated over six soil layers (to 4.6 m depth) following the Richards equation.
       CABLE can be run with interactive biogeochemistry (Wang et al., 2011) and vegetation demography (Haverd et al., 2014),

but both were switched off as leaf area index was prescribed on a per site basis. We applied CABLE to the sites uncalibrated,
       meaning that there was no optimisation of parameters to improve the performance at each individual site. Instead, default
       parameters were taken from the assumed dominant plant functional type (i.e. for savannah ecosystems, CABLE was either run
       as a grass or an evergreen broadleaf tree) at each flux site location. Due to this, these model runs represent a lower bound on the
       possible performance of TBMs at each of these sites. The comparison between the statistical approaches and CABLE provides

insight into the role of underlying site predictability (including ecological memory) in model-observation evaluations.

## 3 Results

### 3.1 Model Performance

The ability of the CC ("current climate"), EM ("environmental memory") and AR ("autoregressive") models to capture the
temporal variability in NEE varied considerably among the 13 OzFlux sites (Figure 1). In general, as sites became drier,

their NEE fluxes became more predictable, independent of the three models used (p-values < 0.02). The introduction of
       lagged memory effects (EM model) consistently improved model performance across sites. The smallest improvement was at
       AU-Tum where the $R^2$ increased from 0.23 to 0.25. AU-GWW experienced the greatest improvement, with memory effects
       increasing $R^2$ from 0.37 to 0.53. The improvement in model performance when introducing memory effects was true across all
       model performance metrics considered, apart from MBE which saw small increases at some sites (see Supplementary Figure

S1). The increase in site predictability associated with environmental memory was also correlated with the MAP at the sites
       (Spearman's $\rho$ = -0.72, p < 0.01). However this relationship between memory and MAP was only significant when all sites
       were considered together and was not apparent when either site grouping (NATT or SAWS) was considered in isolation, albeit
       the sample size is smaller when considering transects seperately (n=13 vs n=6 and 7). Among the savannah sites, the role of lag
       effects increased as the precipitation regime became more seasonal (relative improvement in EM compared to CC, correlated

with the coefficient of variation of precipitation (CVP) at the sites, $\rho$ = 0.97, p < 0.01). By contrast, at the woodland sites, the





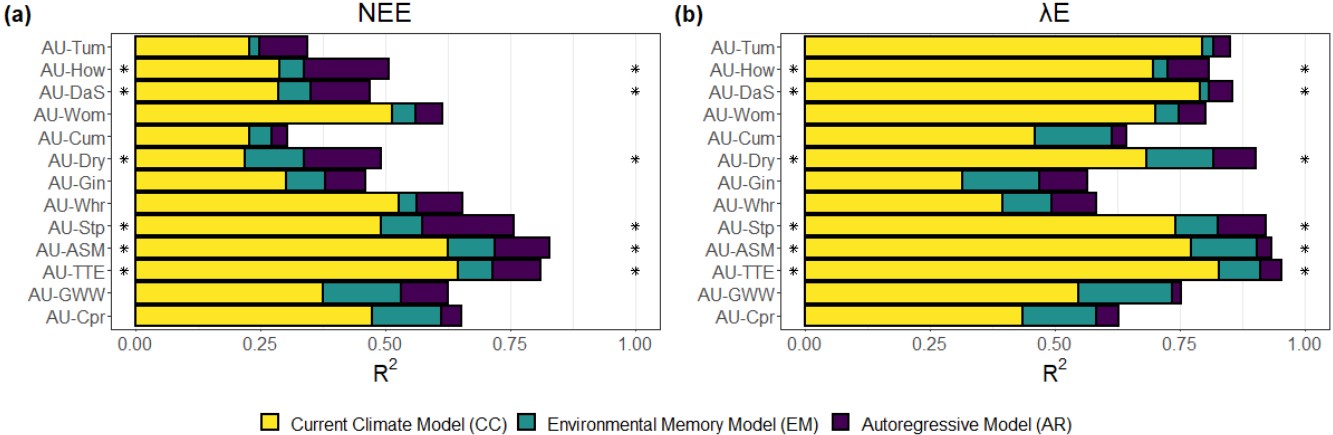

**Figure 1.** Model performance for (a) NEE and (b) $\lambda$E. Sites are on the y-axis, ordered by descending mean annual precipitation with AU-Tum having the highest annual rainfall. The x-axis is the cumulative coefficient of determination, $R^2$. Yellow bars indicate the performance of the current climate-only model (CC model), the turquoise bars are the improvement when memory effects were introduced with the SAM model (EM model) and the purple bar is the performance when the SAM residuals were further modelled with an AR1 process (AR Model). Asterisks indicate sites belonging to the NATT.

importance of memory was instead correlated with mean annual temperature, with hotter sites exhibiting a greater lag influence (relative improvement in EM compared to CC, $\rho = 0.83$, p $< 0.05$).

The remaining predictability captured by the AR model was correlated with the seasonality of precipitation, as expressed by CVP. Introducing the AR1 process had a greater influence on absolute improvement in model performance at sites with greater precipitation variability (purple bar in Figure 1, $\rho = 0.73$, p $< 0.01$). The relative improvement from the AR model compared to the EM model (cf. purple bar relative to the combined yellow and turquoise bars) was also correlated with CVP, but to a lesser extent ($\rho = 0.62$, p = 0.02). When the NATT sites were considered in isolation, this correlation was no longer significant. Instead, there was a strong correlation between the relative improvement from the EM to the AR models and measures of site aridity. Both MAP and WI were correlated with this relative improvement ($\rho = 0.94$, p = 0.02). There were no significant relationships between the AR model performance at SAWS sites and climate metrics.

By contrast to NEE, the $\lambda$E flux was more predictable, with total $R^2$ values once all memory effects were included ranging from 0.56-0.94, compared to $R^2$=0.3-0.82 for NEE (Figure 1b). As with the NEE fluxes, the improvement in $\lambda$E model performance as additional memory components were introduced was maintained across all model performance metrics and sites (see Supplementary Figure S2). The addition of environmental memory improved $R^2$ values at all sites but by varying amounts. AU-DaS saw $R^2$ improve from 0.79 to 0.81 while $R^2$ values for AU-GWW increased from 0.55 to 0.73. The improvement in model performance from the introduction of lagged effects (turquoise bar, Figure 1) was correlated to the dryness of the site. The four wettest sites (MAP > 1000 mm yr$^{-1}$) had an $R^2$ improvement of between 0.02 and 0.05, while the drier sites varied





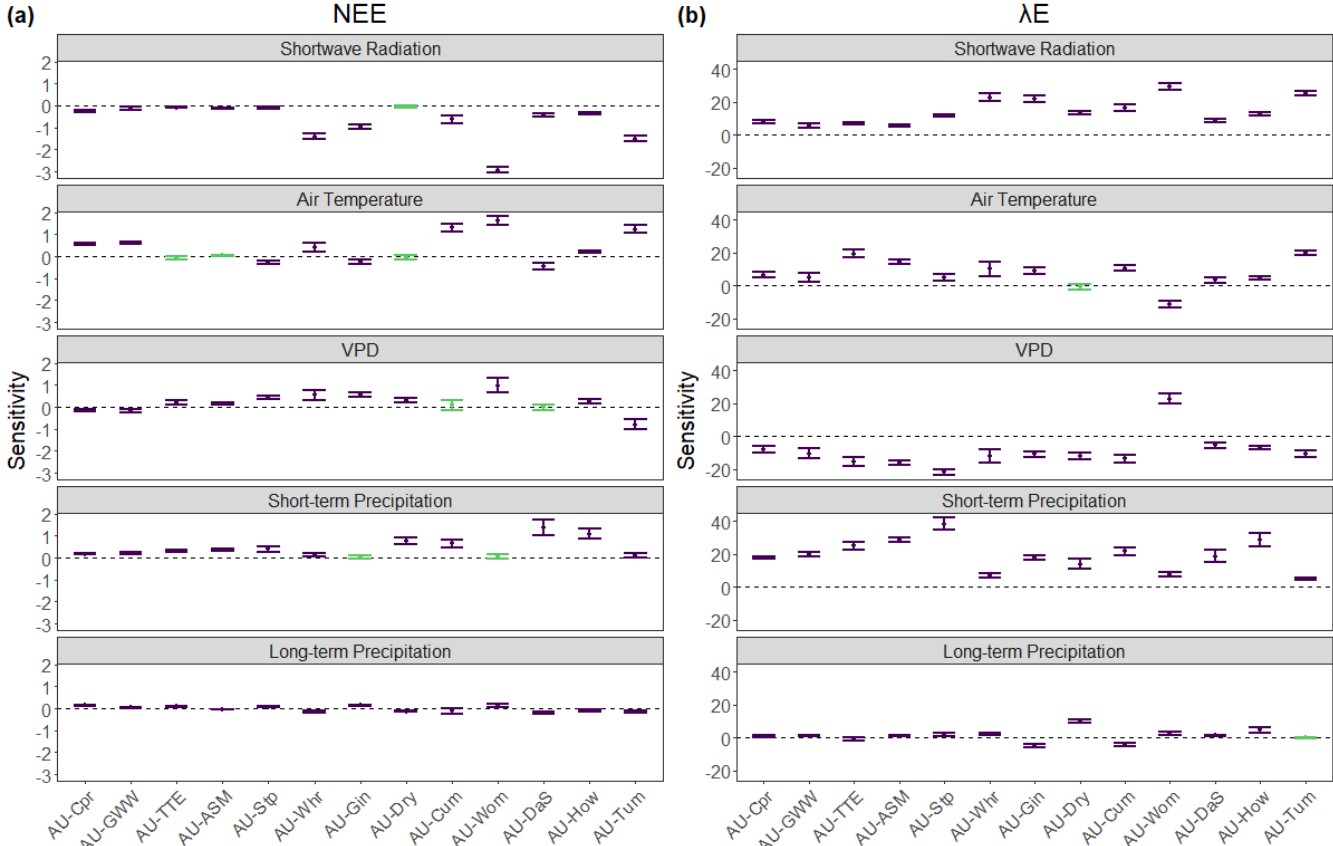

**Figure 2.** Sensitivity of the flux to each climate variable for (a) NEE and (b) λE in the environmental memory models. Sites are on the x-axis, ordered by ascending mean annual precipitation. The y-axis is the sum of all $G_n$ and $D_n$ coefficients from Equation 1 where $n$ includes the climate variable, divided by the mean standard deviation of the corresponding weighted sums from Table 2, such that sensitivity values are comparable between variables. Where the range of sensitivity includes 0, those variables are considered to non-significantly affect the flux and are coloured green in the plots.

from 0.08 to 0.19. Interestingly, the results did not show any clear difference in memory importance between the savannah and woodland sites.

The improvement in performance of the AR1 process for LE fluxes was correlated with the temperature seasonality at the sites (the standard deviation of mean monthly temperatures). This was true for both absolute improvement in $R^2$ and the $R^2$ improvement relative to the EM model ($\rho$ = -0.62, p = 0.03 and $\rho$ = -0.64, p = 0.02, respectively). However, this relationship was not seen in the groups when sites were partitioned into the savannah or woodland subsets.





## 3.2 Sensitivity to Climate Predictors

To explore the impact of individual climate predictors on NEE, the $G_n$ and $D_n$ coefficients from the EM model were summed and normalised by the standard deviation, as shown in Figure 2 (see also Supplemental Figure S3). For NEE (Figure 2a), most climate variables significantly impacted the flux. Air temperature did not significantly affect NEE at three sites, while short-term precipitation and VPD were not significant at two sites each. AU-Dry was the only site where more than one climate variable was not significant. The sensitivity magnitude was generally greater for the short-term variables (past 14 days), than for

long-term precipitation. Shortwave radiation had a mean sensitivity coefficient of -0.69, followed by short-term precipitation (0.44), air temperature (0.37) and VPD (0.18). In comparison, mean NEE sensitivity to long-term precipitation was just -0.017. For the savannah sites, the NEE flux appeared to display greater sensitivity to environmental conditions as sites became wetter, which was most clear for short-term precipitation. The woodland sites had both a wider range, and larger magnitude, of sensitivity to the environmental drivers, with shortwave radiation, air temperature and VPD having a greater impact on NEE

than at the savannah sites.

$\lambda$E fluxes were sensitive to more climate variables than NEE fluxes. Across all sites and variables, only two climate drivers were not significant in their effect on $\lambda$E at a site: air temperature at AU-Dry and long-term precipitation at AU-Tum. The lack of significance in long-term precipitation at AU-Tum may be due to this site being the wettest with a MAP of over 1400 mm yr$^{-1}$ and one of the sites with the most even rainfall distributions (CVP = 32). The driest year in the AU-Tum data is 2006 with

421 mm of precipitation, which is still greater than the MAP at the four driest sites in this study. Another result of interest is that the response of the $\lambda$E flux to air temperature and VPD at AU-Wom, which was the opposite to the relationship at all other sites. AU-Wom exhibited an increase in $\lambda$E with an increase in VPD, and a decrease in the flux as temperature increased.

For both NEE and $\lambda$E fluxes, there was a correlation between the sites' sensitivity to shortwave radiation and their wetness index ($\rho$ = -0.73, p < 0.01 for NEE, $\rho$ = 0.81, p < 0.01 for $\lambda$E). However when this relationship was explored by splitting

between the two vegetation groups, it was only significant for the SAWS sites. The correlation was in fact stronger when only the woodland sites are considered ($\rho$ = -0.86 and 0.86 for NEE and LE respectively, p = 0.02). When the NATT sites were taken in isolation, this relationship between site aridity and sensitivity to shortwave radiation was not apparent.

## 3.3 Timescales of Memory Influence

The cumulative weights from the EM model can provide evidence of the relevant timescales at which the significant climate

variables affected NEE. These are shown for (a) air temperature and (b) long-term precipitation in Figure 3. Following Liu et al. (2019), we assumed that the critical lag timescale is when the cumulative weight reached 0.5, indicated by the dashed line. For air temperature, at sites where this climate metric was significant, all but two sites had a lagged response of > two days. NEE at AU-Stp and AU-Gin had dependence on prior air temperature at longer timescales, around 4-5 days. The timescales at which precipitation affected NEE were much less consistent across sites. AU-DaS had a very short lag of only 21 days while AU-Tum

required over 270 days of prior rainfall to reach a weight of 0.5. The remaining sites had lags to long-term precipitation falling between these extremes but with no obvious correlation between MAP and response timescale. Similarly, there was no clear



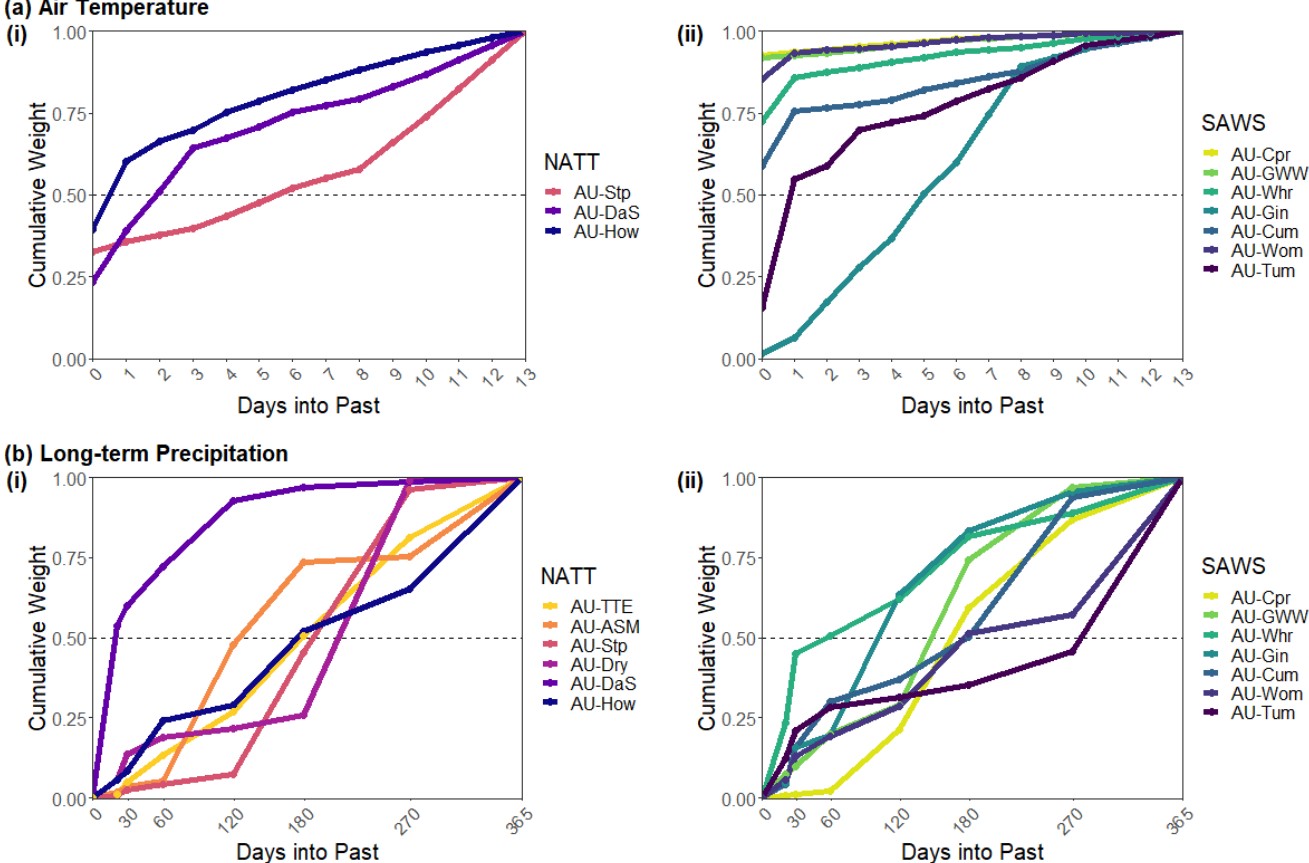

**Figure 3.** Cumulative mean weights from the NEE EM model for (a) air temperature and (b) long-term precipitation. Sites are split into the (i) NATT and (ii) SAWS groups for each climate predictor. Within each group, sites are ordered by mean annual precipitation, with darker colours indicating higher MAP. Only sites where the climate variable is significant in Figure 2(a) are included. The dashed line at a cumulative weight of 0.5 indicates the threshold for the critical lag period. Where the cumulative weights cross this line is considered the timescale of the environmental memory effect for each site for the climatic driver in question.

relationship between lagged responses and the prevailing vegetation at the sites - both NATT and SAWS sites exhibited a range of critical timescales.

Figure 4 shows that the timescale at which the $\lambda E$ flux responded to air temperature is generally longer than that for NEE. 305 This may reflect the contribution of deep soil moisture (and so longer timescales) to transpiration fluxes (driven by VPD associated with higher temperature), whereas the impact of temperature on NEE, via for example heterotrophic respiration, would be controlled by shallower soil moisture (Parton et al., 1988). We found that seven of the twelve sites responded to the air temperature from 3 to 7 days prior. Notably AU-Gin, which had a relatively long response timeframe for NEE to air temperature, had a strong immediate response for $\lambda E$ with the critical timescale occurring at no lag. The shape of the cumulative



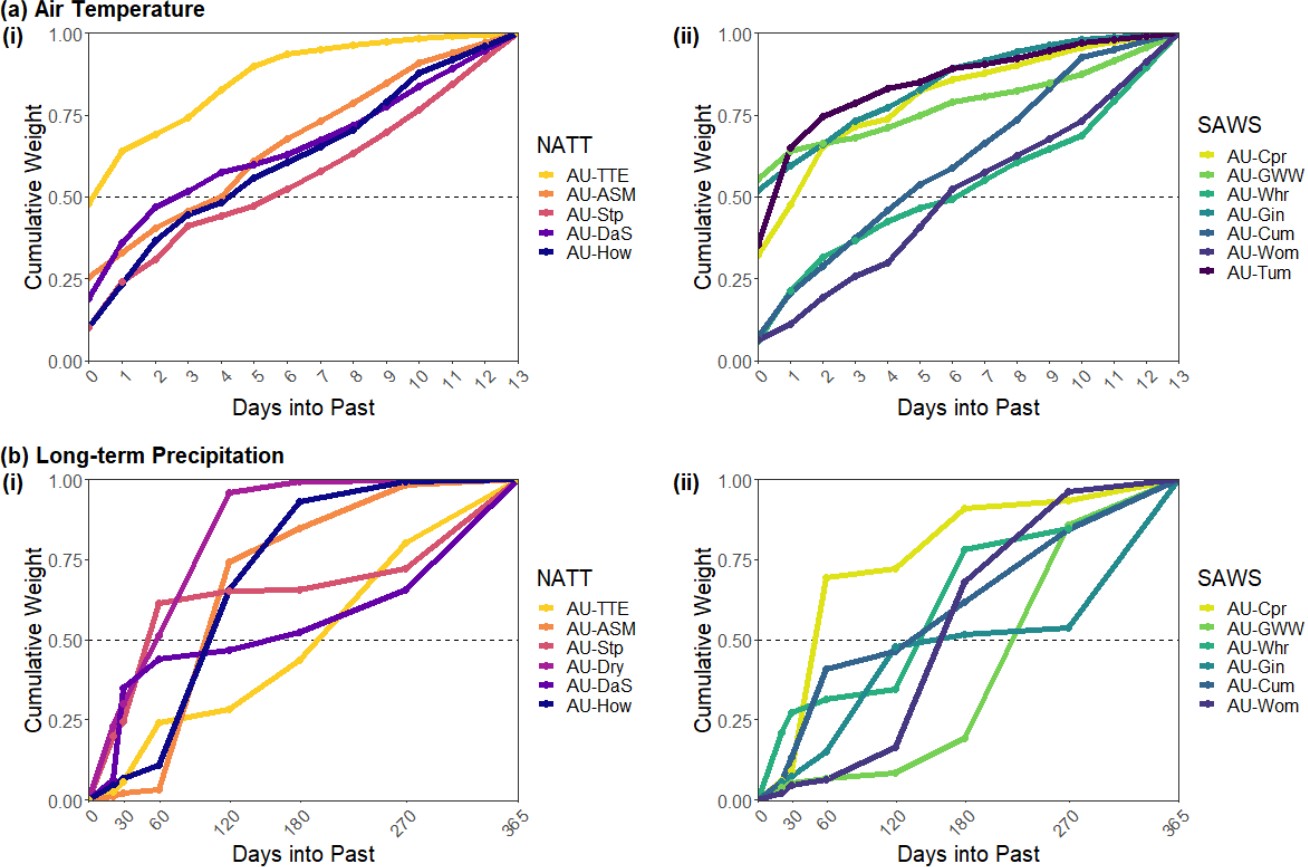

**Figure 4.** Cumulative mean weights from the $\lambda$E EM model for (a) air temperature and (b) long-term precipitation. Sites are split into the (i) NATT and (ii) SAWS groups for each climate predictor. Within each group, sites are ordered by mean annual precipitation, with darker colours indicating higher MAP. Only sites where the climate variable is significant in Figure 2(b) are included. The dashed line at a cumulative weight of 0.5 indicates the threshold for the critical lag period. Where the cumulative weights cross this line is considered the timescale of the environmental memory effect for each site for the climatic driver in question.

weight plots were more similar across sites for $\lambda$E than for NEE, with a consistent increase across each lagged period and a much smaller range of initial weights calculated for the current air temperature.

For long-term precipitation, critical time periods for $\lambda$E ranged from 60 to 270 days. No obvious relationship existed between site aridity and the timescales at which long-term precipitation affected evapotranspiration, indicating that other site characteristics (for example rooting depth) were influencing the lagged effects of rainfall. While the overall range of lagged 315 responses to long-term precipitation was very similar between NEE and $\lambda$E, the critical timescales for each flux differed at most sites (see Figure 3(b) and 4(b)). For instance, at AU-Dry and AU-Stp, the critical lagged timescale for $\lambda$E was 60 days,



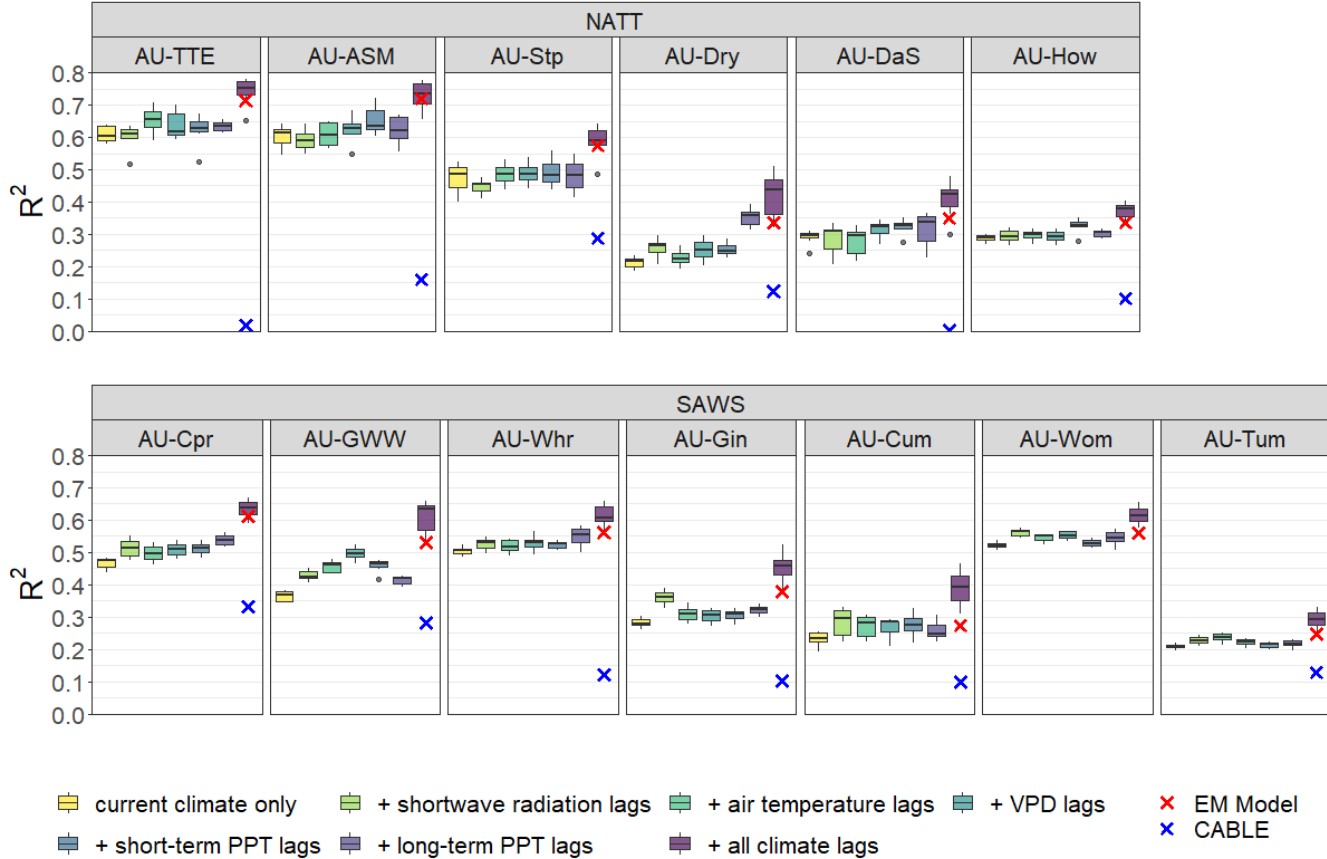

**Figure 5.** $R^2$ values for seven different k-means clustering plus regression models predicting NEE at the flux sites, split by the two vegetation groups. "Current climate" is the k-means model that only considers concurrent environmental observations for predicting NEE. Each of the five lagged environmental predictors are then introduced separately. "+ all climate lags" is the model where every lagged environmental variable is included, which corresponds to the EM SAM model. Boxplots indicate the distribution of the $R^2$ values when these models are run for between two and eight clusters each. The EM SAM model $R^2$ value is indicated by a red cross, and the $R^2$ for an uncalibrated TBM (CABLE) is shown with a blue cross. Note that AU-Wom had no available CABLE output. Sites are ordered from left to right by ascending mean annual precipitation within each vegetation group.

but for NEE a total lag of 270 days was required for a cumulative weight of over 0.5. Conversely, AU-DaS had a short lagged NEE response to long-term precipitation of 20 days but the critical weight was reached at 180 days for λE.

### 3.4 Comparison to Alternative Modelling Approaches

Finally, NEE fluxes were modelled using both a k-means clustering plus regression approach and an uncalibrated TBM, the results of which are shown in Figure 5. At every site, the EM SAM model performed better than a current climate-only k-means





model which is to be expected due to the far greater degrees of freedom in the SAM models. The difference between the median k-means $R^2$ and EM $R^2$ ranges from 0.04 to 0.16. However, the introduction of the lags for individual climate variables to the k-means model generally improved performance, although the EM SAM model outperformed these models in most cases.

When all lags were introduced to the k-means model, performance improved at all sites. Performance was comparable to or better than that of the EM SAM model, confirming the applicability of SAM for predicting terrestrial ecosystem fluxes.

The sites were also modelled using the Australian TBM, the Community Atmosphere Biosphere Land Exchange model (CABLE). In these model runs, CABLE was uncalibrated and so indicates a lower bound on TBM performance at the sites. As shown in Figure 5, this consistently underperformed relative to the predictability expected from the empirical methods. $R^2$

values for the TBM predictions against observed NEE ranged from 0 to 0.33, with a mean of 0.15. Performance was generally better at the SAWS sites than at the NATT sites. CABLE performed better as sites became drier within the SAWS group but this behaviour was not seen for the NATT sites.

## 4 Discussion

Prior to the introduction of the Stochastic Antecedent Modelling framework (Ogle et al., 2015), little work had attempted to

explicitly quantify the role and behaviour of these lagged effects. This paper has provided further evidence that ecosystem fluxes exhibit complicated responses to antecedent conditions and that these responses are important components of ecosystem functioning, and as such need to further explored if these ecosystems are to be properly understood.

### 4.1 To What Extent Does Environmental Memory Matter?

This study has shown that understanding the lagged component in ecosystem responses to climate, that is the environmental

memory, can significantly improve the ability to model site fluxes empirically. Introducing antecedent climate to the NEE model increased the $R^2$ by an average of 0.08, which is a mean relative improvement of 21% in model performance with the relative improvement at individual sites ranging from 8% to 55%. For $\lambda E$, the improvement ranged from 2% to 49%, with a mean of 19%. This is reflective of the generally higher predictability of the $\lambda E$ flux compared to NEE, with more of the variance in the flux explained by current climate only. This improvement in model performance for both fluxes indicates that

exploring the role of environmental memory further could substantially improve our understanding of site functioning.

While the EM model results provide a direct indication of the role of antecedent climate, we further modelled the flux residuals with an autoregressive lag-1 model (the AR model). The ensuing $R^2$ value obtained was interpreted as a lower bound on overall site predictability based on prior conditions, contingent on the structural assumption of a lag of one time step. This is because the AR model captures remaining predictability from the previous day's fluxes without identifying the source of this

influence. For instance, this dependence on prior day NEE may be affected by seasonal leaf area, potential site disturbances or even more unique impacts, such as insect infestation. However, it could also be representing other environmental drivers that we have not explicitly accounted for.





By characterising the extent of individual site memory statistically, we hope to stimulate future site measurement campaigns, hypothesis development that examines what drives memory variability, and ultimately guide model development. Although eddy covariance analysis is a mature field, relatively little work has been done on quantifying environmental memory. Instead, the focus has been on immediate site responses to disturbance events and meteorological extremes (Ciais et al., 2005; von Buttlar et al., 2018; Teuling et al., 2010; Flach et al., 2018). At the site level, further examination of other controls on carbon and water fluxes (e.g. variability in leaf area, root-zone soil moisture, the contribution of non-transpiration components, and the role of non-structural carbohydrates) may unlock explanations for why environmental memory varies both across the precipitation gradients, but also amongst similar geographic sites (e.g. AU-ASM and AU-TTE, which are closely located). For example, we found that our capacity to simulate NEE fluxes at AU-Cum was seemingly poorly explained by current climate. It is notable that Peters et al. 2021 GCB recently highlighted the high tolerance of drought stress (xylem embolism resistance, $p_50$: -4.07 to -5.82 MPa) of species at AU-Cum, which may explain an apparent decoupling between carbon fluxes and current meteorological conditions. Pinning down the exact mechanistic explanation will remain for future work, but our results motivate the search for hypotheses to explain differences in site behaviour.

### 4.2 The benefits of Stochastic Antecedent Modelling

Through the application of Stochastic Antecedent Modelling, we have been able to identify the importance of environmental memory in ecosystem fluxes at a diverse range of Australian sites. Unlike traditional methods of exploring antecedent effects, SAM makes few prior assumptions about critical lag lengths beyond a prescribed maximum lag of interest (i.e., the 14 day window we assigned to the short-term variables). The Bayesian framework also yields the full (joint) posterior distribution for the parameters of interest, from which we can compute summaries such as credible intervals, allowing a critical assessment of the relative significance of various memory drivers. However, the SAM method is computationally intensive and requires long data records which may reduce its potential applications. In this paper, we also performed similar analysis using the basic machine-learning approach of k-means clustering plus regression. The results from the k-means modelling were broadly consistent with those from the SAM approach. Both modelling methodologies produced predictions with similar $R^2$ values (5) and saw improved performance when lagged climate effects were introduced. The k-means modelling in this study has provided a novel, independent check on the suitability and performance of the SAM approach, and the consistent results increase our confidence in the findings from the SAM model. Importantly, while computation of k-means was significantly faster than SAM, the k-means approach lacks the inherent interpretability of SAM. While SAM is explicitly designed to infer the sensitivity to predictors and the timescales at which lags exist, the k-means clustering plus regression approach would require further work and modelling to fully explore these aspects of flux responses, which is beyond the scope of this study.

This study also showed how the relative influence of different drivers of NEE and λE fluxes can be discerned. By normalising the sensitivity parameters for each environmental driver, we further the work of Liu et al. (2019), allowing the individual climate predictors to be compared between variables and across sites. This increases the information provided by the SAM approach regarding the site behaviour and is a further argument for its use to explore the timescales of ecosystem response. Liu et al. (2019) provided a strong argument that including the influence of environmental memory is key for predicting ecosystem





fluxes. Our results suggest that more careful site evaluation is likely to be required to truly understand the impact and source of these memory effects. Fortunately, the SAM approach is well-positioned to provide the necessary insights to shine a new light on site dynamics and memory influence.

### 390  4.3    The Importance of Across-Site Heterogeneity

One of the key conclusions from Liu et al. (2019) was that as sites become more arid, the importance of antecedent effects increases. However, Liu et al. (2019) considered 42 sites from across the globe, incorporating a wide range of biomes and species. As such, there is potential for confounding factors to be influencing the importance of environmental memory at each site. This study reduces some of this uncertainty by limiting its scope to 13 sites, all located within Australia. This means that,

as well as limiting the diversity of species and climates studied, a greater understanding of each individual site is possible. Similarly to Liu et al. (2019), these sites were grouped by biome, although we only had two groups - savannahs/grasslands within the NATT and woodlands in the SAWS group. Each biome group contains sites with a range of MAP and WI values. When these sites are viewed together, the importance of memory is strongly correlated with site aridity (improvement in $R^2$ between CC and EM models, $\rho$ = -0.72, p-value < 0.01), consistent with the conclusions of Liu et al. (2019). However, when

the sites are split by our vegetation groupings, this significant correlation is only seen for the SAWS group ($\rho$ = -0.86, p-value < 0.05), again noting that total sample size is reduced when we split by vegetation groupd. The NATT sites had no correlation between site memory and aridity ($\rho$ = -0.49, p-value = 0.36), despite having a very strong rainfall gradient. Note that both groups have ranges of aridity that include sites spanning from "arid" to "humid" with the WI at NATT sites between 0.12 and 0.75 and at SAWS sites between 0.11 and 1.2 (Trabucco and Zomer, 2018). This result indicates that grouping many sites

together to explore relationships based on a single metric can obscure more nuanced understanding of the processes involved, or the key site characteristics driving such relationships. For instance, at the savannah sites, we found that NEE sensitivity to short-term precipitation increases as site MAP increases. We hypothesise this increased sensitivity is driven by the monsoonal nature of rainfall at the NATT sites with greater MAP. Our results also show that the relationship between shortwave radiation and site aridity is only seen in the SAWS grouping, not at the NATT sites. This is potentially due to the greater proportion of

woody vegetation in SAWS sites resulting in a greater rooting depth and less frequent water stress. As such, these sites are likely to be energy-limited and hence transpiration is linked to days of high photosynthetically active radiation.

We have also found an inverse response of $\lambda$E to temperature and VPD at AU-Wom to all other sites. It is not clear what the driver is here and it is unlikely to be related to the correlation between temperature and VPD, as this is higher at the other sites which did not exhibit this behaviour ($\rho$ = 0.85 at AU-Wom, six other sites have higher $\rho$ values, up to 0.92).

One possible cause is that AU-Wom generally has the lowest VPD observations across all sites with only AU-Tum showing similar but greater values (median VPD = 0.20 kPa, other sites range from 0.31 to 1.94 kPa). While other sites may tend to close their stomata, limiting transpiration as VPD increases, AU-Wom is potentially below the VPD threshold at which stomata begin to limit transpiration. Similarly, Griebel et al. (2020) found that AU-Wom does not limit transpiration during hot temperatures and heatwaves, potentially due to access to deep water reserves. However, the inverse response could also be due

to the vegetation composition at AU-Wom, with the potential for this effect to have been caused by photosynthetic inhibition



at high temperatures (and high VPD), which would further explain why this opposite response is not seen in the NEE flux. The prevailing wind direction, which is seasonally dependent, heavily influences the flux tower footprint and affects the climate conditions at AU-Wom (Griebel et al., 2016). This could potentially contribute to the inverse behaviour seen, if changes in VPD are correlated with significant changes in the vegetated area being measured by the tower. Additional exploration of the
lags experienced at the AU-Wom site is probably necessary if this response is to be more precisely attributed.

Many studies have also highlighted how global relationships do not hold at regional-, local- or even site-level (Knapp and Smith, 2001; Knapp et al., 2017; Lauenroth and Sala, 1992; Ukkola et al., 2021; Wilcox et al., 2016). Such non-transferability is related to the issue of spatial versus temporal relationships, and particularly the "vegetation structure constraint" (Lauenroth and Sala, 1992). This is where, due to the slow timescales at which species composition and plant function respond to changes
in climate, individual sites are unable to fully utilise any inter-annual variability in climate conditions (Lauenroth and Sala, 1992). Here, we have shown that, within the range of Australian ecosystems analysed, environmental memory is not clearly related to site aridity for savannah sites. In comparison, among woodland sites, the link between environmental memory and site aridity appears to be stronger. Our results point to a need to better understand the role of individual site characteristics (i.e. root-zone water access), in determining predictability of carbon and water fluxes.

**4.4 Implications for TBM Evaluations**

Flux data is routinely used to benchmark and improve TBM performance (Abramowitz, 2012; Abramowitz et al., 2008; Best et al., 2015; Haughton et al., 2016, 2018b; Nearing et al., 2018). In spite of this, relatively few studies have proposed that assessments of TBM performance at flux sites should also consider the underlying site predictability, but see (Haughton et al., 2018a). Here, we argue that the confounding effect of baseline predictability is essential when comparing models that may
have been tested at different sites and also in determining which ecosystems a model can best represent. We suggest that models tested at more arid sites, where we report higher predictability both with and without memory taken into account, should be expected to perform better than models tested at those sites which exhibit a lower baseline predictability of fluxes. Such consideration of individual site behaviour could affect reported results from models. For instance, Whitley et al. (2016) found that, when modelling GPP and $\lambda$E fluxes, the performance of a suite of TBMs improved slightly across the NATT
sites as the MAP decreased. This is consistent with our findings of higher predictability at more arid sites. Importantly, this performance could be reinterpreted given the baseline predictability calculated using the SAM approach. Across the sites included in both this paper and Whitley et al., they found model-averaged correlation coefficients for $\lambda$E predictions had a maximum difference between sites of 0.08, ranging from 0.56 (AU-Dry) to 0.64 (AU-Stp). Our analysis found a similar difference between correlation coefficients of 0.06 (0.85 at AU-How and 0.91 at AU-Stp). This might show that both empirical
models and TBMs perform better at drier sites. However, when we consider the TBM performance relative to the empirical SAM performance, AU-How actually performs better (TBMs capture 70% of the expected predictability assumed from the SAM model) than the drier AU-Stp site (where TBMs capture 66%).

Similarly, Barraza et al. (2017) modelled $\lambda$E across the NATT using various indices of surface conductance and found that the sites with greater Eucalyptus occurrence (i.e. the wetter sites: AU-How, AU-DaS and AU-Dry) were better represented by

their model. However, if the reported $R^2$ values from this study are considered relative to the values found in this study, then their models perform even better at the wet sites compared to the dry. The Eucalyptus-dominated sites have $R^2$ values very similar to those from this study while at AU-Stp and AU-ASM (the drier sites), the $R^2$ values found in Barraza et al. (2017) are substantially lower.

Similarly, the CABLE model results in Figure 5 illustrate how the results from empirical models can be used as a baseline

for TBM performance. At AU-Cum, AU-Dry, AU-Gin, and AU-Whr, the $R^2$ from CABLE is close to 0.1. However, from the SAM and k-means approaches, it is apparent that AU-Whr is more predictable than the other three sites. The performance of CABLE at AU-Whr is only around 18% of the empirical models. At AU-Cum, the $R^2$ of CABLE is approximately 25% and 37% of the $R^2$ of k-means and SAM respectively. Hence, while initially CABLE appears to perform similarly well at both sites, it can be seen that the TBM is capturing more of the expected predictability at AU-Cum than at AU-Whr. As such, the

baseline predictability of sites as calculated from detailed empirical models such as ours clearly provide a framework by which we can reinterpret the predictability of ecosystems and hence how well TBMs are performing.

## 5   Conclusions

Accurate prediction of carbon and water fluxes is key to understanding the role that terrestrial ecosystems will play in a changing climate. This study built on previous work utilising Stochastic Antecedent Modelling to provide further evidence

that environmental memory is a key component of both net ecosystem exchange and latent heat fluxes. In general, the role of this memory effect increases as sites become more arid, yet we have shown that this relationship is confounded by individual site characteristics/behaviour. By separating the influence of various predictors on NEE and $\lambda$E fluxes, it becomes clear that despite this broad scale relationship with aridity, very different mechanisms are at play across sites. The differences we report in site behaviour should motivate a range of new hypotheses in future research to understand the controls on variability in

predictability of site fluxes. Finally, we argue that a consideration of both site predictability and environmental memory should form a key part of terrestrial biosphere model evaluation and future process development.

*Code and data availability.*   OzFlux data is available from ozflux.org.au. Model codes, example workflows and select analysis scripts are available at https://github.com/JDCP93/OzFlux_SAM

*Author contributions.*   JCP performed all model runs and code analysis, with input from MDK, GA, YL and KO. JCP wrote the manuscript

with substantial input from MDK, GA, MJH, JC, YL and KO. AJP and NHN provided further input on the manuscript.

*Competing interests.*   MDK is a member of the editorial board of Biogeosciences. The peer-review process was guided by an independent editor, and the authors have also no other competing interests to declare.



*Acknowledgements.* Jon Cranko Page, Martin G. De Kauwe and Gab Abramowitz acknowledge support from the Australian Research Council (ARC) Centre of Excellence for Climate Extremes (CE170100023)





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
