# Peer review of "Examining the Role of Environmental Memory in the Predictability of Carbon and Water Fluxes Across Australian Ecosystems"

_Biogeosciences, 2021_

## Author Response (AR1)

**RESPONSE TO REVIEWER 1**

We thank the reviewer for their constructive comments, and we address their various concerns below.

Referee comments are highlighted in blue, with our response below in black in each case.

*In this manuscript, Page et al. apply the stochastic antecedent modeling framework to predict fluxes of carbon and water across a number of Australian flux tower sites. They found that consideration of lagged meteorological effects significantly improved the prediction of fluxes, and this improvement was greatest at arid sites. This result was replicated across a number of different modeling approaches that consider different ways of accounting for these lags.*

*The use of methods to understand the influence of temporal lags on ecosystem fluxes is a very important way to improve our understanding of ecosystems and guide the development of broader models such as TBMs, and the authors do an excellent job of highlighting this. The writing throughout the manuscript is incredibly clear, as are the analyses. As a result, I have very few technical comments. Great job!*

Thank you very much for your positive summary of our manuscript.

*I do have a few broader concerns, largely surrounding the precise contributions this manuscript is trying to make to the field.*

- *As far as I can tell, the methods and results are very similar to that of Liu et al. 2019, which the authors cite extensively. I think that a stronger case needs to be made for what new contribution is being made here. I agree that the paper makes a compelling case that lagged effects matter for improving flux predictions, but does not make much of a distinction for how this manuscript adds on to the Liu et al. work, or Samuels-Crow et al. 2020, or the other cited papers using the SAM model (or other lagged models) to predict fluxes. The primary novelty seems to be that they are using sites that are more closely distributed than Liu et al., which the authors claim can help bypass some of the confounding factors brought up in the Liu et al. paper. This may be true to some extent, but the sites in this manuscript are also quite spaced apart, and in my view is it just speculation that having a different geographic distribution of sites improves the interpretability of the results.*

Thank you for highlighting this issue. The study certainly builds upon Liu et al. (2019) and we agree that a stronger argument for the manuscript's novelty is required.

We make four novel contributions in this manuscript:

1. We evaluate our approach on latent heat and evaluate how legacy affects both NEE and latent heat fluxes. Liu et al. only explored NEE. As our results show, the interpretation for both predictability and ecosystem legacy differ markedly by flux, with latent heat considerably more predictable (Figure 1)
2. We compare the SAM model to another empirical approach, which constitutes a robust, independent assessment of SAM (not in either Liu or Samuels-Crow et al.).

3. We demonstrate how to normalise the sensitivity coefficients, and hence allow comparison between environmental drivers. This is a significant step forward from Liu et al. (2019).
4. Finally, we make an important framing argument around the need to first understand "ecosystem baseline predictability" for TBM assessment, a point rarely considered in most flux model intercomparisons.

We tend to disagree with the reviewer's point about site selection not affecting the interpretability of results. Liu et al. (2019) reviewed 42 sites located across the US, Europe, Australia, Russia and the Amazon, which cover 9 IGBP classifications (CSH, DBF, EBF, ENF, GRA, MF, OSH, SAV, WSA). Our sites cover only 4 IGBP classifications (EBF, GRA, SAV, WSA), on the same continent, while keeping a similar rainfall gradient (Liu et al, 320 – 1651 mm y$^{-1}$ with an outlier at 3041 mm y$^{-1}$; our study 256 – 1491 mm y$^{-1}$). As such, we have reduced the confounding impact of vegetation type (and continent, which includes various environmental factors) while maintaining the aridity scale of the study. This is further reduced by focussing predominately on a single species – for instance, the woody vegetation at 9 of our sites is dominated by Eucalypt species. The NATT in particular was specifically deployed to allow for examination of savannah functioning over an aridity gradient, and it has been suggested that spatial patterns of flux here are dominated by structural changes as opposed to vegetation species differences (Hutley et al., 2011). As a consequence, we are able for example to hypothesise reasons for the low memory impact at AU-Cum and the inverse response to VPD at AU-Wom. However, we accept both the benefit of this could be better explained, and that further exploration can be performed here. We have hopefully addressed these shortcomings below.

Changes include:

L9 – We added a sentence to the abstract that highlights the novelty of the study:

"By focussing our analysis on a single continent (and predominately on single genus), we reduced the degrees of variation between each site, providing a novel chance to explore the unique characteristics that might drive the importance of memory."

L85 – We added a sentence to explicitly mention the NATT sites as designed for this kind of study. This highlights the novelty of applying an existing method to these specific sites, insofar as the reduction in confounding factors:

"We include the sites of the North Australian Tropical Transect (NATT) as an explicit case study, since this "living laboratory" covers a steep rainfall gradient without a correspondingly strong change in vegetation (Hutley et al., 2011)."

L86 – We included an additional sentence to highlight the inclusion of latent heat and the purpose of this:

"By applying the SAM framework to λE in addition to NEE, we explore how the timescales of response vary between these coupled fluxes which can improve our understanding of the processes involved in environmental memory."

L95-100 – We have added the IGBP classifications of the sites so that the increase in homogeneity in this study is provided further evidence.

"These vary from tropical grasslands to semi-arid shrublands and savannahs **(IGBP classifications of grassland, savannah and woody savannah)** along a steep rainfall gradient (312 to 1486 mm annual precipitation) running from north to south in the Northern Territory, Australia. Secondly, we grouped the Southern Australian Woodland Sites (SAWS). These were selected as sites with a greater proportion of woody vegetation than the NATT sites **(IGBP classifications of savannah, woody savannah and evergreen broadleaf forest)**…"

L376 – This has been split into two sentences that should reinforce the benefit of the k-means modelling in improving confidence in the SAM results:

"The k-means modelling in this study has provided a novel, independent check on the suitability and performance of the SAM approach. The consistent results increase our confidence in the findings from the SAM model **and reduces the likelihood that our findings are influenced by the structural assumptions of the SAM model.**"

L381 – We have included an additional sentence here which helps to explain the importance of the k-means approach we used in comparison to SAM:

"Since the k-means clustering plus regression often outperformed the SAM model, we have identified that the SAM approach does not provide an upper bound on the information available from the flux data. As such, our results highlight the need to explore the role of environmental memory using different approaches, including use of alternative machine learning techniques."

L388 – We also changed "the SAM approach is well-positioned…" to "the SAM approach, **together with other machine learning techniques,** is well-positioned…" which further reinforces our suggestion that multiple empirical models should be used in parallel to explore memory effects.

- *While I do think that methods such as these are important tools for showing that lagged processes matter in TBMs, the discussion does not seem to offer much targeted recommendation besides claiming lagged processes matter (which is not novel). Do the results suggest any specific ways that lags could or should be incorporated? Which processes/drivers could feasibly be incorporated into TBMs? On L354 there is text about how the results could guide model development… how?*

This is a legitimate comment, which we appreciate you highlighting. Unfortunately, identifying all of the precise processes involved in our results was beyond the scope of this current piece of work. We have suggested some important features that may explain the observed behaviour (e.g. VPD at AU-Wom, (xylem embolism resistance at AU-Cum), but the truth is that the current observations do not allow us to further distinguish why behaviour differs across sites. Instead, we see our work as being motivation, it should facilitate further studies to explain the behaviour we see, likely requiring further site instrumentation e.g. data from the new critical zone observatory network (Monitoring Australia's life-sustaining 'Critical Zone' resources, 2021).

In revisions, we have taken the sentence mentioned from L354 and used this as the introduction instead for a further paragraph at the end of section 4.1. This highlights some proposed next steps in identifying the processes to be included in TBMs:

"By characterising the extent of individual site memory statistically, we hope to stimulate future site measurement campaigns, hypothesis development that examines what drives memory variability, and ultimately guide model development. As for which TBM modules would need to be adjusted to fully capture environmental memory, this approach needs to be applied at many more individual sites. This would allow us to identify functional relationships to a greater extent. However, such application needs to carefully pursued, using not just SAM but other machine learning approaches (such as the k-means clustering plus regression as we have demonstrated), to ensure that any results are process-based and not just structural assumptions from the use of a single modelling approach. By combining multiple empirical studies of environmental memory, we can understand the key lags that aid prediction of ecosystem fluxes and how these vary across site characteristics."

- *I completely see the idea behind including CABLE results, but in my opinion they do not offer much by themselves. It seems obvious that these complex lagged effect models can predict fluxes better than CABLE. For one, they are statistical models run on the actual data and not process-based models. CABLE was also not calibrated, so the actual model performance is unknown. Is there any idea how a fully calibrated CABLE might perform compared to the lagged effects models?*

Yes, there is, and indeed we could motive this better. To address this, we have amended the paragraph beginning on L216 to:

"For further comparison, we also consider the performance of an uncalibrated TBM in simulating site NEE. The TBM used was the CSIRO Atmosphere Biosphere Land Exchange (CABLE) model (Kowalczyk et al., 2006), a land surface scheme that can be run offline with prescribed meteorological forcing (De Kauwe et al., 2015b; Decker et al., 2017; Haverd et al., 2018; Ukkola et al., 2016b; Wang et al., 2011), or fully coupled (Lorenz et al., 2014; Pitman et al., 2011) within the Australian Community Climate Earth System Simulator (ACCESS; Kowalczyk et al. 2013v). CABLE models the exchange of carbon, energy and water fluxes at the land surface, representing the vegetation with a single layer, two- leaf (sunlit/shaded) canopy model (Wang and Leuning, 1998) and a detailed treatment of within-canopy turbulence (Raupach, 1994; Raupach et al., 1997). Soil water and heat conduction are numerically integrated over six soil layers (to 4.6 m depth) following the Richards equation. CABLE can be run with interactive biogeochemistry (Wang et al., 2011) and vegetation demography (Haverd et al., 2014), but both were switched off as leaf area index was prescribed on a per site basis. **CABLE is a state of the art TBM that performs similarly to other TBMs used in global coupled modelling (Best et al., 2015).** We applied CABLE to the sites uncalibrated, meaning that there was no optimisation of parameters to improve the performance at each individual site. Instead, default parameters were taken from the assumed dominant plant functional type (i.e. for savannah ecosystems, CABLE was either run as a grass or an evergreen broadleaf tree) at each flux site location. **CABLE's reported performance at the 13 sites in this study is then essentially the performance one might expect if CABLE were run in a global coupled model – unlike the empirical models it is being compared to, it is not calibrated with site data, so in some sense this is not a fair comparison. Nevertheless, there are strong indicators that local calibration of TBMs offers relatively minor performance increases (i.e. that structural inadequacies remain), and that empirical approaches benefit to a much greater**

**degree by the inclusion of local calibration information (Abramowitz et al., 2007). There is also compelling evidence that TBMs share biases (Haughton et al., 2016). We suggest therefore that this comparison should highlight how much more appropriate empirical approaches are for investigating ecosystem memory effects than TBMs with additional parametrisations, where existing structural inadequacies in TBMs could cloud the interpretation of the inclusion of lagged effects. Effectively,** these **CABLE** model runs represent a lower bound on the possible performance of TBMs at each of these sites **and so the** comparison between the statistical approaches and CABLE provides insight into the role of underlying site predictability (including environmental memory) in model-observation evaluations."

*Line by line comments*

*L2-3: This sentence is a bit confusing, are direct versus long-term physiological responses supposed to be in contrast to each other?*

We have changed the sentence to:

> "Constraining climate-carbon cycle feedbacks requires improving our understanding of **both the immediate** and long-term plant physiological responses to climate."

This clarifies that we are talking about the different timescales at which plants react to environmental conditions.

*L11-14: Would there be an easy way to quickly describe what type of memory effects helped improve model performance? Specificity could help here.*

We agree with this comment and have modified the sentence to specify that the memory is represented as the lagged antecedent drivers. As such we changed the sentence from "both fluxes were more predictable when memory effects were included in the model" to "both fluxes were more predictable when memory effects (expressed as lagged climate predictors) were included in the model."

*L57: Probably most accurate to just say non-structural carbohydrates here, since 'storage' implies longer time scales than the likely lag between photosynthesis and synthesis of structural carbon.*

The reference to "storage" was removed and replaced by "non-structural carbohydrates" as suggested.

*L109: How were they predicted? Is this a typo? It is confusing to mention anything regarding prediction here.*

Here we had used "predicted" to refer to the modelling of fluxes using the climate predictors. We have changed this to "modelled" for clarity.

*L121: Where were PET data from?*

The PET data is calculated by Trabucco and Zomer (2018) from the WorldClim dataset. We have added two further citations here (Zomer et al., 2007; Zomer et al., 2008) to clarify the source of this data.

*L146: Are these short-term predictors half-hourly? Hourly? What time scale?*

Line 116 has been amended to make it clear that the observations are collected at a half-hourly resolution and were aggregated to daily measurements. It has become:

> "All OzFlux data were extracted at a **half-hourly** timestep, **aggregated to daily data,** screened to only include complete calendar years and then mean-centred."

We also have modified L146 to

> "$CLIMATE_n(t)$ is a weighted sum of **daily** climate **measurements**".

*L151: Similarly, is long-term precipitation always 1 year? It is unclear from the text, which says "up to a year prior". Is this summed precipitation, or averaged?*

We have changed this sentence to

> "Long-term precipitation (**mean rainfall calculated over varying periods**, up to 365 days prior, **Table 3**) is included…".

This should draw the reader's attention to Table 3 which provides the timeblocks that prior year precipitation was split into it.

*L390-411: I think the other possibility – that the trend just doesn't hold when considering smaller sample sizes – should be given equal value here. This text could also be shortened since it is a fairly minor result in my opinion, but I will just mention that as a suggestion.*

As suggested, we have slightly condensed this paragraph and included further reference to the potential impacts of a smaller sample size (where previously, this was only mentioned in the Results section). We believe that the initial section of this paragraph is important, since a key aim of the paper was to reduce site variance by considering just a single continent. As such, this was kept but the additional two examples of "lost correlation" were removed and additional sentences added to discuss sample size. Combined, the paragraph has changed to:

> "One of the key conclusions from Liu et al. (2019) was that as sites become more arid, the importance of antecedent effects increases. However, Liu et al. (2019) considered 42 sites from across the globe, incorporating a wide range of biomes and species. As such, there is potential for confounding factors to be influencing the importance of environmental memory at each site. This study reduces some of this uncertainty by limiting its scope to 13 sites, all located within Australia. This means that, as well as limiting the diversity of species and climates studied, a greater understanding of each individual site is possible. Similarly to Liu et al. (2019), these sites were grouped by biome, although we only had two groups - savannahs/grasslands within the NATT and woodlands in the SAWS group. Each biome group contains sites with a range of MAP and WI values. When these sites are viewed together, the importance of memory is strongly correlated with site aridity

(improvement in R2 between CC and EM models, $\rho = -0.72$, p-value $< 0.01$), consistent with the conclusions of Liu et al. (2019). In contrast, when the sites are split by our vegetation groupings, this significant correlation is only seen for the SAWS group ($\rho = -0.86$, p-value $< 0.05$). The NATT sites had no correlation between site memory and aridity ($\rho = -0.49$, p-value $= 0.36$), despite having a very strong rainfall gradient. Note that both groups have ranges of aridity that include sites spanning from "arid" to "humid" with the WI at NATT sites between 0.12 and 0.75 and at SAWS sites between 0.11 and 1.2 (Trabucco and Zomer, 2018). This result **potentially** indicates that grouping many sites together to explore relationships based on a single metric can obscure more nuanced understanding of the processes involved, or the key site characteristics driving such relationships. **However, this loss of correlation when grouping by vegetation type could simply be an artefact of the smaller sample sizes. By decreasing from a sample of 13 sites to 6 or 7, it is more likely for erroneous relationships to be identified (or not)."**

*L401: Typo, "groupd".*

Corrected.

*L440-442: This is a pretty big claim, especially considering that the basis for it is just from a few arid sites in one region. Is there any reason to believe that model performance in arid regions is better globally (or even outside of Australia)? The improvement you see could be related to model structure or data quality or other factors, as opposed to just aridity.*

This is a very valid criticism and we have taken it on board. Since we argue that aridity is not a strong indicator of site predictability, this statement was at odds with the rest of the manuscript. As such, the sentence has been changed as follows to better reflect our intentions regarding how studies such as ours should guide TBM development and evaluation:

> "We suggest that models tested at more arid sites, where we report higher predictability both with and without memory taken into account, should be expected to perform better than models tested at those sites which exhibit a lower baseline predictability of fluxes."

has become

> "Our results indicate that process-based TBMs tested at sites that exhibit greater predictability from simple empirical models, such as SAM or k-means as used in this paper, might be expected to perform better than TBMs tested at those sites which exhibit a lower baseline predictability of fluxes."

*Figure 5. The caption makes it seem like the "+ all climate lags" is equivalent to the EM model, which in my understanding, it is not. Plus, the red crosses differ from the "+ all climate lags" boxplots, which makes that wording confusing.*

You are correct and this was poorly worded. The caption has been amended to more accurately reflect that the EM model and "+ climate lags" k-means model are equivalent only insofar as they are driven by exactly the same predictors.

""+climate lags" is the model where every lagged environmental variable is included, which corresponds to the EM SAM model." has been changed to ""+ all climate lags" is the model where every lagged environmental variable is included, and hence utilises exactly the same predictors as the EM SAM model."

We thank the reviewer for their helpful and encouraging comments, and we address their various concerns below. Referee comments are in blue, with our response below highlighted in black in each case.

The manuscript "Examining the Role of Environmental Memory in the Predictability of Carbon and Water Fluxes Across Australian Ecosystems" by Cranko Page and coauthors deals with the crucial topic of ecological memory and how this affects biosphere-atmosphere net CO2 fluxes and latent heat.

The manuscript builds on Liu et al., 2018, but it's incremental because of the use of different models (i.e., k-mean clustering) and the use of terrestrial biosphere models.

The manuscript is well written. The analysis is well done and robust. The results are fascinating and well discussed. I think this will article provide an exciting contribution, and it will be of great interest for the Biogeoscience journal readership. It's an excellent manuscript.

I found that only one aspect should be improved: a deeper discussion on how the community can modify models to describe the ecological memory better. I found this aspect a bit weak and can be improved.

For the rest, I believe that the manuscript is well done, enjoyable, and scientifically very robust.

Thank you for your positive summary of our manuscript.

I have a few comments I suggest the authors should address:

Line 4: I suggest including what is meant for structural lags: "…structural lags (i.e., …)"

We agree with this comment, especially as we previously provided examples of extreme events earlier in the sentence. As such, we have amended this to read "structural lags (such as delays between rainfall and peak plant water content, or between a precipitation deficit and down-regulation of productivity)".

Line 5: I suggest substituting models with "terrestrial biosphere models" to be specific

Agreed. This has been added.

Line 10: substitute latent hear with lambda

This has been changed as suggested.

Line 45: In my opinion, this statement critically depends on the timing of the precipitation with respect to the phenological stage. Please consider rephrasing to account for this comment

We agree with the Reviewer, particularly for deciduous grasses and trees; however, for evergreen species, the phenological stage has less relevance. Our original text was deliberately general and so we have opted not to change it.

Line 51: I agree on the vegetation type, but it would be better to be more specific on which aspects and differences between vegetation types can be critical confounding factors (different allocation strategies, height, etc). This would set the ground for the discussion.

We agree that "vegetation type" was vague and have changed this sentence to highlight a few of the specific reasons for vegetation type affecting the timescales of response:

"Confounding factors, such as differing vegetation characteristics (including the proportion of woody vegetation, rooting depth and varying allocation strategies), prevailing climate, interacting processes, and prior extreme events, all influence the magnitude and timescale of these lags and their impact on ecosystem fluxes."

Lines 63: "models fail to capture the impact of water stored in reservoirs with longer response times to climate". I think the authors should clarify why models do not describe well the climate impact: missing water table depth, poor description of soil layers and the root profile, etc.

We have added a sentence here to clarify some of the reasons for the failure of models to capture this impact:

"Such model failures may in part be due to incorrect rooting depths, poor soil profile characterisation, or a lack of representation of these long-term storage pools such as groundwater or wetlands."

Table 1: I suggest adding the vegetation type

Thank you for this suggestion. We have added the ecoregion classification for each site to the table. This provides further support to our grouping of sites into two different vegetation groups. The caption for Table 1 has been updated with the following sentence to address the new column:

"World Ecoregion is the biome classification of each site which is based on climatic regime and ecological structure, among other criteria (Olson et al., 2001; Beringer et al., 2016)."

Line 103: If I am not wrong latent heat and net ecosystem exchange are already defined

This is true, and there were further instances where these acronyms were either defined again or used prior to being defined. As such, the below changes have been made:

Line 42: "NEE" has been replaced with "net ecosystem exchange (NEE)" as the first definition

Line 77: "Net ecosystem exchange (NEE)" has been changed to "NEE"

Line 103: Full names replaced with acronyms

Line 105: Specify already here the time scale used (i.e., daily data). This information is reported a few paragraphs later. I think it will help the clarity. Also, the assumption of a certain degree of independence between NEE and latent heat would not be valid at hourly data during the daytime, when photosynthesis dominates the signal of NEE, so better to clarify that you are talking about daily data.

Line 103 has become "In this analysis we used observations of daily NEE and $\lambda E$ fluxes"

Line 115: I see the importance of using the long-term and consistent NDVI data. The authors use NDVI as a phenological and structural proxy. I suggest at least discussing the use of radar data rather than only optical data. For instance, I invite the authors to test Sentinel-1 data rather than MODIS NDVI for a few selected sites or give a perspective beyond the NDVI.

NDVI was used as it represents a widely used proxy for phenology. In our study, the NDVI data allows us to identify the site's sensitivity to each climate driver based on site greenness.

The reviewer makes an interesting suggesting to use Sentinel-1 data; however, as this was launched in 2013, it does not cover our full site record. Moreover, it is worth noting that satellite-based products that work well in other ecosystems/environments often have weaker skill in Australian environments (e.g. Leuning et al., 2005). By contrast, the NDVI product is well tested and understood (e.g. Rifai et al., 2021), which helps statistical interpretation (confidence) of our model results.

We have plans to extend our approach and factoring in multiple remote sensing datasets, so we may revisit the suggested Sentinel data as part of this work, as assessing new satellite data can be an entire piece of research.

Would you please specify which temporal resolution (8-day composite?)? I think it might be a piece of important information for the study of lag effects.

The NDVI data used is daily values calculated from a 16-day period of observations. The full algorithm for the calculation of this composite is available at the cited source. On line 117 we added the following sentence to clarify that we are taking the daily data available from the cited 16-day composite dataset:

"This dataset is calculated at a daily timestep based on a 16-day composite of observations."

The methodology is sounding. I suggest calculating linear relaxed precipitation instead of using the 15-days rainfall average. Linear relaxed precipitation can be calculated with a backward moving weighted average with 15 days width and weights that linearly decrease along with the window. The result is a pseudo soil moisture time series that might serve the purpose of the analysis better than the 15-days rainfall average.

We thank the reviewer for this suggestion – it's indeed a good idea. However, such a change is not necessary, since a mechanism to achieve this temporal weighting already exists within the Stochastic Antecedent Modelling framework. The weights in our 14-day antecedent sums are inferred from the data. If a linear decrease in weights within this window is suggested by the data, then this would be seen in the results. This is indeed the case for some sites – see for instance the short-term precipitation timescale at AU-Cpr in Figure S5 where the logarithmic-like shape of the plot indicates decreasing weight with increasing lag. We also note that some

sites do not follow the same timescales, which implies that the response is not universally linear and therefore may not accurately be captured by the reviewer's proposed method.

Finally, the authors assume a Laplace distribution of the error at the daily time scale. There are contrasting reports in the literature, and some suggest that even at half-hourly time-scale, the Laplace distribution is the result of the the superimposition of two gaussian distributions of daytime and nighttime error (e.g. Lasslop et al., 2008), and at daily time scale the error is likely to be gaussian. I don't think this would impact the results, but I think it should be clarified to report all the positions in the literature.

This is a very pertinent point, although we do note that the Laplace distribution still performs well for these fluxes as per Lasslop et al. (2008). We have clarified that differing results are present in the literature by amending line 130 to the below:

"While the exact distribution of flux errors is site-dependent and can vary between super-imposed Gaussian distributions (Lasslop et al., 2008) or Student's t-distribution (Weber et al., 2018), daily NEE is assumed to be Laplace-distributed (Richardson et al., 2006) with mean $\mu NEE$ and variance $\sigma^2$ in line with Liu et al. (2019)"

Figure 2 and results section on the sensitivity: The sensitivities should have units if I understood the method. Would you please add them everywhere?

Thank you for this correction. The relevant units have been added where required (including Figure S3 in the supplement).

Line 195: Something missing at the end of the sentence?

We believe the sentences in this paragraph to be complete.

Line 303: "No clear relationship between lagged responses and the prevailing vegetation at the sites.."

It would be interesting to check that relationship with quantitative vegetation characteristics reported for many OzFlux sites rather than a general statement on the prevailing vegetation. Did the author verify if the average canopy height, C4 fraction, fractional tree cover, rooting depth if known, etc., are possible controlling factors of the lagged response?

Our statement was unclear and has been amended: "No clear relationship between lagged response timescales and the prevalence of woody vegetation at the sites…".

The reviewer notes a range of potential controlling factors, but data on these is not easily obtained (e.g., C4 fraction, which is typically model-derived; rooting depth, which is rarely well resolved across sites). One of our hopes is that our findings provide fresh inquiries into trying to understand site behaviours, which will include collection of information like that suggested by the reviewer.

Line 350-351: I agree, but I think other potential causes are: 1) lags between respiration and photosynthesis due to transport or relocation of CO2 and assimilated (Mencuccini, M. and Hölttä 2010), as well as 2) potential rapid dynamic modulation of allocation due to phenology or response to stress.

We accept these suggestions for other potential causes and have amended line 351 to include the following:

"For instance, this dependence on prior day flux may be affected by seasonal leaf area, delays between photosynthesis and respiration (Mencuccini and Hölttä, 2010), or potential site disturbances. It could also be representing environmental drivers that we have not explicitly accounted for, lagged allocation, or more unique impacts such as insect infestation."

Line 360: please check the reference

This has been corrected.

Line 416: typo in 1:94 kPa

This now reads "1.94 kPa" as intended.

Figure 3) and 4). There are double labels for some panels. I find it confusing. I suggest to change it.

We appreciate the concern for the clarity of these labels but believe them to be necessary. The plots need to be split by both environmental driver and vegetation group, as otherwise the data become too hard to parse from the plots. We use the Latin alphabet to identify the driver and Roman numerals to identify the vegetation group, together with an explanation in the captions. Both of these remain consistent across Figure 3 and 4 and so we believe the figures are unambiguous.

Discussion on the terrestrial biosphere models: as mentioned above, I think there is a need to describe better processes that can be improved in models to improve the predictability of the fluxes

We thank the reviewer for this comment and agree that, in general, it is an important knowledge gap to better describe the responsible processes. However, to capture environmental memory within TBMs, two separate tasks are required: a) identify and quantify the timescales and b) link these to a process or series of modelled processes. We believe that this study falls within the scope of task a) and adds to the growing body of recent work that is highlighting the timescales at which terrestrial ecosystems respond to the environment (for example, see Bastos et al., 2020; Ciais et al., 2005; Feldman et al., 2020; Liu et al., 2018; Ogle et al., 2015). These results have identified and quantified some of the critical timescales of response, although our understanding of environmental memory is still incomplete - especially when linking these timescales to processes. Identifying the responsible process (or series of) will require model-hypothesis testing against data. While the available flux data could provide a good starting point for this, experimental work that captures step changes in climate or recovery from extremes are likely needed. This is an exciting direction for future research and our future projects may target this more directly.

We have added the below at line 467:

"Our results add to a growing body of research (for example, see Bastos et al., 2020; Ciais et al., 2005; Feldman et al., 2020; Liu et al., 2018; Ogle et al., 2015) that identifies an important role of

ecosystem "memory" in the terrestrial fluxes. This first step, including the characterisation of the timescale of influence, the processes affected (e.g., LE vs NEE, etc), the controlling environmental driver and site-to-site variability, is critical to improving TBMs. It is widely acknowledged that capturing legacy processes in TBMs is important (e.g., acclimation, recovery from climate extremes, link between carbon uptake and growth, canopy defoliation, etc.), but to develop the theory, we first need a strong evidence base against which we can probe model predictions. The challenge now is to link the statistical findings to mechanisms and then demonstrate that capturing these processes in models leads to improvements in site predictions. This second step will require applications of our approach (or similar) to both field and targeted experimental data, with progress likely to be made by linking directly to model-hypothesis testing (e.g. Katul et al., 2001; Mahecha et al., 2010)."

**RESPONSE TO EDITOR**

Thank you again for submitting your work to Biogeosciences. As you saw, the reviewers mostly had favorable comments about your work, although Reviewer 1 had some significant comments. Thank you for addressing these comments in detail. However, your response to Reviewer regarding the inclusion of uncalibrated CABLE is not fully convincing. If the SAM models perform better than CABLE/TBMs, wouldn't an *upper* bound of the CABLE performance (e.g. with calibration) be the more useful comparison? It is understandable that calibrating CABLE may be a lot of effort, but if the authors really believe there would be minimal gain because "there are strong indicators that local calibration of TBMs offers relatively minor performance increases (i.e. that structural inadequacies remain)" these indicators should be explained and contextualized in more detail. Please consider clarifying or amending this comment when uploading your revision files.

We thank the Editor for raising this issue of calibration and allowing us the opportunity to further develop our response.

To better support our argument, we have now calibrated CABLE at three OzFlux sites, by varying the two dominant physiological parameters (Vcmax - the maximum carboxylation rate & g1 - the slope of the sensitivity of stomatal conductance to photosynthesis). As shown in Figure C1 below, the result of calibration is an overall shift in magnitude of the simulated fluxes (i.e. a vertical shift); however, it has very little impact on the skill of the simulated flux (the temporal behaviour). Any improvement in the temporal behaviour is likely to simply be a coincident result of changes in water availability resulting from the shift in overall magnitude (i.e. a higher or lower Vcmax/g1).

Alternatively, one could attempt to calibrate the parameterisation of the root-zone, as this may affect the temporal dynamics. In a recent paper (Mu et al., 2021) this was attempted for CABLE and the authors demonstrated that: (a) substantial soil moisture data to depth (> 4m) was required (which we do not have in this study); and (b) that calibrated improvements, while possible, require a very careful and detailed investigation - far beyond the scope of this paper (if one ignores the lack of soil moisture data to underpin such an undertaking).

Overall, we hope Figure C1 better demonstrates our point about "structural inadequacies remaining" - we believe that what our analysis does is shine a light on process gaps that control behaviour, rather than a role for improved calibration. We believe these further model runs, together with the prior comments to Reviewer 1, illustrate that calibrating CABLE would not drastically alter our discussion.

In addition, it is worth noting that the comparison to "default" CABLE should not really be viewed as "uncalibrated". This is because the application of CABLE at a point-scale makes use of the model's parameterisation by plant functional type, which in many instances relies on these parameters being informed by data (i.e. De Kauwe et al., 2015). In revision, we have been careful to better make this point to avoid the reader confusing the application with an uncalibrated model, as per the changes outlined below.

Further, the inclusion of CABLE in the paper is largely utilised to illustrate how empirical methods such as SAM can provide information on the relative performance of TBMs. We discuss how, relative only to itself, the version of CABLE we run can be argued to perform better at some sites than others e.g. AU-Cum rather than AU-Whr. We do not discuss the merits of CABLE against other TBMs, and only briefly compare CABLE directly with the SAM approach (on Line 329).

Changes to the manuscript:

Line 216: We have removed the word "uncalibrated"

Line 225: The sentence has been changed to "At each site, we applied CABLE with the parameterisation taken from the assumed dominant plant functional type (PFT) at the flux tower location". The following sentence from the original draft then clarifies that CABLE is not optimised with site data.

Figure 5 Caption: We have replaced the word "uncalibrated" with "PFT-parametrised".

Line 320: This has been amended to "Finally, NEE fluxes were modelled using both a k-means clustering plus regression approach and the CABLE TBM parametrised by PFT, the results of which are shown in Figure 5."

Line 328: This has been amended to "In these model runs, CABLE was parametrised by assumed PFT, with no site-specific calibration, and so indicates a lower bound on TBM performance at the sites."

While revising the manuscript, we discovered an issue with our model runs for the Ti Tree East site (AU-TTE). As such, we have taken the decision to remove this site from our analysis and so the study now covers 12 sites rather than 13. All results and analysis have

been performed again, with the conclusions being effectively indistinguishable between the original submission and the amended 12 site study. Minor changes to correlation coefficients were found that did not affect the significance of all but two reported relationships. The manuscript has been updated to include the new correlation coefficients and p-values. The more substantial changes are detailed below:

Line 253: The relationship between AR relative improvement at NATT sites and site aridity was no longer significant. As such, these two sentences were removed.

Line 292: The relationship between temperature seasonality and the AR performance for λE was no longer significant. This paragraph has been changed to "The improvement in performance of the AR1 process for λE fluxes was not correlated with any of the climate metrics under consideration. This was true both when all sites were considered at once and when sites were partitioned into the savannah or woodland subsets."

[Figure]

Figure C1: Time series of observed NEE (black) and NEE modelled by CABLE. In panels of (a), Vcmax was fixed at 30 while g1 varies from 2 to 7. In (b), g1 was fixed at 5 while Vcmax varies. Three sites are shown to illustrate the consistent change in magnitude of the flux with minor temporal differences.

**Bibliography**

Bastos, A., Ciais, P., Friedlingstein, P., Sitch, S., Pongratz, J., Fan, L., Wigneron, J.P., Weber, U., Reichstein, M., Fu, Z., Anthoni, P., Arneth, A., Haverd, V., Jain, A.K., Joetzjer, E., Knauer, J., Lienert, S., Loughran, T., McGuire, P.C., Tian, H., Viovy, N., Zaehle, S., 2020. Direct and seasonal legacy effects of the 2018 heat wave and drought on European ecosystem productivity. Sci. Adv. 6, eaba2724. https://doi.org/10.1126/sciadv.aba2724

Ciais, P., Reichstein, M., Viovy, N., Granier, A., Ogée, J., Allard, V., Aubinet, M., Buchmann, N., Bernhofer, C., Carrara, A., Chevallier, F., De Noblet, N., Friend, A.D., Friedlingstein, P., Grünwald, T., Heinesch, B., Keronen, P., Knohl, A., Krinner, G., Loustau, D., Manca, G., Matteucci, G., Miglietta, F., Ourcival, J.M., Papale, D., Pilegaard, K., Rambal, S., Seufert, G., Soussana, J.F., Sanz, M.J., Schulze, E.D., Vesala, T., Valentini, R., 2005. Europe-wide reduction in primary productivity caused by the heat and drought in 2003. Nature 437, 529–533. https://doi.org/10.1038/nature03972

De Kauwe, M.G., Kala, J., Lin, Y.-S., Pitman, A.J., Medlyn, B.E., Duursma, R.A., Abramowitz, G., Wang, Y.-P., Miralles, D.G., 2015. A test of an optimal stomatal conductance scheme within the CABLE land surface model. Geosci. Model Dev. 8, 431–452. https://doi.org/10.5194/gmd-8-431-2015

Feldman, A.F., Short Gianotti, D.J., Konings, A.G., Gentine, P., Entekhabi, D., 2020. Patterns of plant rehydration and growth following pulses of soil moisture availability. Biogeosciences Discussions 1–24. https://doi.org/10.5194/bg-2020-380

Hutley, L.B., Beringer, J., Isaac, P.R., Hacker, J.M., Cernusak, L.A., 2011. A sub-continental scale living laboratory: Spatial patterns of savanna vegetation over a rainfall gradient in northern Australia. Agricultural and Forest Meteorology, Savanna Patterns of Energy and Carbon Integrated Across the Landscape (SPECIAL) 151, 1417–1428. https://doi.org/10.1016/j.agrformet.2011.03.002

Leuning, R., Cleugh, H.A., Zegelin, S.J., Hughes, D., 2005. Carbon and water fluxes over a temperate Eucalyptus forest and a tropical wet/dry savanna in Australia: measurements and comparison with MODIS remote sensing estimates. Agricultural and Forest Meteorology 129, 151–173. https://doi.org/10.1016/j.agrformet.2004.12.004

Liu, L., Zhang, Y., Wu, S., Li, S., Qin, D., 2018. Water memory effects and their impacts on global vegetation productivity and resilience. Scientific Reports 8, 2962. https://doi.org/10.1038/s41598-018-21339-4

Monitoring Australia's life-sustaining 'Critical Zone' resources, 2021. TERN Australia, viewed 8 October 2021, https://www.tern.org.au/news-ozczo-announcement

Mu, M., De Kauwe, M.G., Ukkola, A.M., Pitman, A.J., Gimeno, T.E., Medlyn, B.E., Or, D., Yang, J., Ellsworth, D.S., 2021. Evaluating a land surface model at a water-limited site: implications for land surface contributions to droughts and heatwaves. Hydrology and Earth System Sciences 25, 447–471. https://doi.org/10.5194/hess-25-447-2021

Ogle, K., Barber, J.J., Barron-Gafford, G.A., Bentley, L.P., Young, J.M., Huxman, T.E., Loik, M.E., Tissue, D.T., 2015. Quantifying ecological memory in plant and ecosystem processes. Ecol Lett 18, 221–235. https://doi.org/10.1111/ele.12399

Rifai, S.W., De Kauwe, M.G., Ukkola, A.M., Cernusak, L.A., Meir, P., Medlyn, B.E., Pitman, A.J., 2021. Thirty-eight years of CO2 fertilization have outpaced growing aridity to drive greening of

Australian woody ecosystems. Biogeosciences Discussions 1–41. https://doi.org/10.5194/bg-2021-218

Trabucco, A., Zomer, R.J., 2018. Global Aridity Index and Potential Evapo-Transpiration (ET0) Climate Database v2. CGIAR Consortium for Spatial Information (CGIAR-CSI).

Zomer, R., Bossio, D., Trabucco, A., Yuanjie, L., Gupta, D., Singh, V., 2007. Trees and water: smallholder agroforestry on irrigated lands in Northern India, IWMI research report. Colombo, Sri Lanka.

Zomer, R., Trabucco, A., Bossio, D., Verchot, L., 2008. Climate Change Mitigation: A Spatial Analysis of Global Land Suitability for Clean Development Mechanism Afforestation and Reforestation. Agriculture Ecosystems & Environment 126, 67–80. https://doi.org/10.1016/j.agee.2008.01.014